# AutoGO: Automated Computation Graph Optimization for Neural Network Evolution

**Mohammad Salameh**[1][*], **Keith G. Mills**[1,2][*][†], **Negar Hassanpour**[1], **Fred X. Han**[1],
**Shuting Zhang**[3], **Wei Lu**[1], **Shangling Jui**[3], **Chunhua Zhou**[3], **Fengyu Sun**[3], **Di Niu**[2]

[1]Huawei Technologies Canada. [2]Dept. ECE, University of Alberta. [3]Huawei Kirin Solution, China.
{mohammad.salameh, negar.hassanpour2, fred.xuefei.han1,
jui.shangling, zhouchunhua}@huawei.com,
{kgmills, dniu}@ualberta.ca, {zhangshuting8, robin.luwei, sunfengyu}@hisilicon.com

## Abstract

Optimizing Deep Neural Networks (DNNs) to obtain high-quality models for efficient real-world deployment has posed multi-faceted challenges to machine learning engineers. Existing methods either search for neural architectures in heuristic design spaces or apply low-level adjustments to computation primitives to improve inference efficiency on hardware. We present Automated Graph Optimization (AutoGO), a framework to evolve neural networks in a low-level Computation Graph (CG) of primitive operations to improve both its performance and hardware friendliness. Through a tokenization scheme, AutoGO performs variable-sized segment mutations, making both primitive changes and larger-grained changes to CGs. We introduce our segmentation and mutation algorithms, efficient frequent segment mining technique, as well as a pretrained context-aware predictor to estimate the impact of segment replacements. Extensive experimental results show that AutoGO can automatically evolve several typical large convolutional networks to achieve significant task performance improvement and FLOPs reduction on a range of CV tasks, ranging from Classification, Semantic Segmentation, Human Pose Estimation, to Super Resolution, yet without introducing any newer primitive operations. We also demonstrate the lightweight deployment results of AutoGO-optimized super-resolution and denoising U-Nets on a cycle simulator for a Neural Processing Unit (NPU), achieving PSNR improvement and latency/power reduction simultaneously. Code available at https://github.com/Ascend-Research/AutoGO.

## 1 Introduction

Deep Neural Networks (DNNs) have achieved great success in Computer Vision (CV) and Natural Language Processing (NLP) tasks. A major trend toward achieving better performance on benchmarks is adopting large and computationally demanding deep models [7]. However, successful and efficient deployment of deep neural networks onto diverse and specific hardware devices, including neural processing units on the edge, significantly hinges upon engineering proper neural architectures that are both excellent in task performance while meeting hardware friendliness objectives.

A range of techniques have been proposed by academia and industry to solve the hardware-friendly deployment challenges of DNNs [46]. Neural Architecture Search (NAS) replaces the manual design process of DNNs, achieving remarkable performance in several applications in CV [67, 11, 9, 6] and NLP [32, 10, 12]. While NAS can utilize a flexible range of search algorithms [53, 11, 79, 47], the search space adopted by NAS is based on heuristics, either searching for an optimal macro-net

---

[*]Equal contribution.
[†]Work done during an internship at Huawei.

37th Conference on Neural Information Processing Systems (NeurIPS 2023).

construction based on predefined blocks, e.g., MBConv blocks in MobileNets [59, 26], or stacking searchable cells by heuristic rules [39, 17, 73]. These heuristic rules may not be efficient to the target hardware device for deployment and may limit the potential gain from NAS methods. On the other hand, graph rewriting [70, 30] operates on the tensor computation graph of a DNN to improve its inference efficiency on hardware. Rewriting involves applying a set of subgraph substitution rules that preserve mathematical functionality of the original DNN, which does not alter or reduce the neural architecture to achieve better task performance or fitness to hardware.

In this paper, we propose Automated Graph Optimization (AutoGO), a generic graph optimization framework to evolve a given neural architecture for efficient and low-power deployment onto a specific hardware device. Unlike traditional NAS which builds networks from scratch in a heuristic search space or from hand-crafted building blocks [39, 63, 15], AutoGO enhances both hardware-friendliness and task performance of a DNN, by evolving its underlying Computation Graph (CG) using computational units composed of operations extracted from NAS benchmarks. AutoGO automatically improves typical neural networks in terms of benchmark performance on several CV tasks ranging from classification, semantic segmentation, to super-resolution without relying on newer operations. It also automates lightweight DNN deployment onto mobile neural processing units while preserving task performance, hence replacing manual tweaking efforts by ML engineers. We present the following key contributions in designing the AutoGO framework:

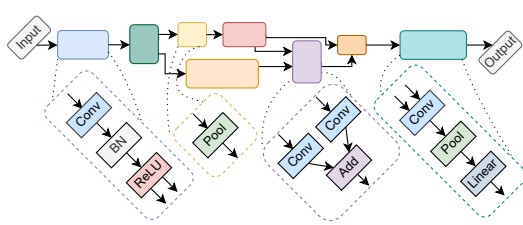

Figure 1: A DNN can be partitioned into disjoint subgraphs (segments). Segments contain a variable number of inputs, outputs, and nodes ranging from primitive operations to complex subgraphs.

AutoGO optimizes DNNs on a Computation Graph (CG) of low-level primitives extracted from TensorFlow [1] models, which allow us to optimize all types of operations and their hyperparameters like filter size and latent tensor dimensions and thus offer a holistic fine-grained view of DNNs. Unlike graph rewriting which preserves mathematical equivalence, AutoGO alters the CG of a DNN for task performance and fitness to hardware.

Rather than relying on predefined blocks, the basic units for mutation in AutoGO are computation subgraphs, which we call segments, as illustrated in Figure 1. Segments are mined from a large number of CGs from several NAS benchmarks based on frequent subgraph mining in a data-driven manner. We use Byte Pair Encoding (BPE), an efficient tokenization technique [20] from NLP, to segment CGs and merge frequent operations into segments. By extracting and including segments of variable sizes into our database, AutoGO enables both primitive operation changes and larger-scaled block changes to a DNN.

AutoGO leverages an evolutionary algorithm to mutate our BPE-segmented source network. The segment mutation process is guided by hardware friendliness metrics and a pre-trained neural predictor to estimate the performance of the mutant network resulting from segment replacement. We propose a neural predictor which explicitly considers the positional and contextual information of a segment replacement made in a CG and directly models the performance gain. Mutations are also coupled with a resolution propagation scheme that solves for the tensor shapes in the replacement segment to ensure architectural validity.

Extensive experiments demonstrate that AutoGO can enhance the performance of the best architectures in several public architecture benchmarks, e.g., NAS-Bench-101 [71], NAS-Bench-201 [17], HiAML, Inception, and Two-Path [48]. Additionally, AutoGO can automatically optimize several typical large CNN architectures, including ResNets [24], VGG [61], and EDSR [38] on a breadth of CV tasks including Classification, Semantic Segmentation, Human Pose Estimation, and Super Resolution. We show that AutoGO can improve their performance while making them lightweight, without using newer generations of operations that do not appear in the original network. Finally, to demonstrate the real-world applicability of our framework, we show results of AutoGO-optimized FSRCNN [16] (for super-resolution) and image denoising U-Net [55] for low-power or low-latency deployment using a cycle simulator for a Huawei mobile Neural Processing Unit (NPU).

## 2 Related Work

**NAS Benchmarks and Neural Predictors.** NAS-Benchmarks comprise architectures from a given search space and their accuracy performance. NAS-Bench-101 [71] and NAS-Bench-201 [17] provide the performance of 423k and 15.6k architectures, respectively, on CIFAR-10. Benchmarks enable the rapid development of search algorithms and neural predictors. Neural predictors [43, 54, 41, 73, 65, 40] treat NAS benchmarks as datasets and learn to estimate the performance of architectures in a given search space, and thus constitute a low-cost avenue for performance evaluation.

However, NAS benchmarks suffer from several drawbacks. First, benchmarks only provide performance annotations for architectures inside a manually designed fixed search space. Thus, any tweaks to decrease FLOPs or latency beyond the search space requires training the new architecture from scratch. Second, existing NAS Benchmarks are mostly cell-based [39] and compose a network by stacking the same cell structure multiple times. This structure forms a high-level architecture representation that is insensitive to spatial details such as latent tensor dimensions, which vary along the depth of the network and influence hardware-friendliness [49]. As most existing neural predictors learn using high-level cell representations, these drawbacks hamper their deployment generalizability. In contrast, AutoGO can mutate an architecture beyond its original, manually-defined design space by utilizing a low-level, spatially-sensitive representation.

**Computational Graphs for DNN Hardware Friendliness.** Multiple subfields explore how to reduce the carbon footprint and time cost of DNNs. Pruning and quantization methods [36] aim to reduce the number of parameters or lower the bit precision of model weights, respectively. Graph rewriting methods like TASO [30] and TENSAT [70] consider mathematically equivalent substitutions, e.g., merging or splitting parallel convolutions and applying the associative/distributive properties. These schemes usually require spatial details like resolution and channel size to perform rewrites. Hence, they represent DNNs using Computation Graphs (CG) [50, 22], which treat each primitive operation as a node and use the network forward pass to define edge connectivity. Similarly, AutoGO also uses a low-level CG representation. But different from these approaches, it aims to evolve the architecture of an untrained DNN to improve performance while also optimizing hardware friendliness.

**Neural Architecture Design Space.** Several works employ NAS over large spaces by jointly searching over macro and micro-structures for both block type and tensor dimensions [64, 15]. Human expertise is at the core of these design choices to constrain the search for high-performing architectures. [57] model a search space as a 3-level hierarchical graph to overcome the reliance on expert knowledge. [13] propose Neural Search Space Evolution, which progressively grows a current search space by adding unseen operation candidates. Unlike the above work, we do not limit ourselves to a pre-designed skeleton with spatial or topological constraints at any network position. Rather, we incorporate search space and architectural knowledge into a neural predictor. Also, instead of manually defining the search units [17, 59, 62, 42], we mine these units from NAS benchmarks. In particular, we utilize Frequent Subgraph Mining (FSM) [31] to discover interesting and frequent patterns in the computation graphs in a data-driven way. FSM requires conducting expensive steps when graphs are large such as extracting patterns, inspecting isomorphism, and checking if subgraphs are frequent enough to be considered interesting. Algorithms [68, 69, 27] trade result completeness and accuracy for efficiency to overcome run time and memory inefficiency [19]. In NAS, [56] utilize Weisfeiler-Lehman (WL) graph kernel to extract useful network features but only applies it shallow cell-based DAG structure of NAS-Bench-201. In contrast, we propose an efficient approach to mine frequent subgraphs by converting CGs to sequences and applying BPE [20] to extract subsequences, which produces a fixed-size vocabulary of subgraphs of varying sizes.

## 3 AutoGO: Automated Computation Graph Optimization

The AutoGO framework operates on the Computation Graph (CG) of an input DNN architecture extracted from the in-memory graph structure of a `tensorflow.keras` model or `.pb` file. CGs are directed acyclic graphs (DAGs), where nodes represent primitive operations that are indecomposable computation operations in deep learning frameworks like ONNX [5] and PyTorch [52], e.g., Convolutions, Pooling, ReLU, Add, etc., while edges represent forward-passes between operations. Specifically, node features include operation type, input and output tensor resolution dimensions and weight tensor shape if applicable.

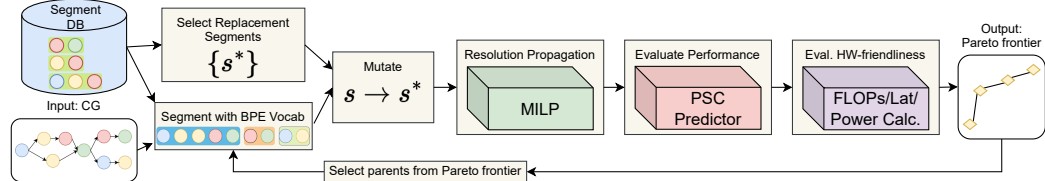

Figure 2: AutoGO takes the CG of a neural network as input and improves it using an algorithm-mined segment database and pre-trained mutation performance predictor.

Figure 2 provides an overview of the proposed AutoGO operating on the CG level. AutoGO mutates the CG of an input DNN by utilizing a database of *segments* (Sec. 3.1), which are frequent subgraphs mined from a variety of NAS-Benchmarks via an efficient tokenization method. A Pareto front evolution strategy (Sec. 3.2) performs segment mutations to the CG, while using resolution propagation to ensure network validity. Finally, a pretrained Predecessor-Segment-suCcessor (PSC) neural predictor (Sec. 3.3) together with selected hardware metrics will guide the evolution by assessing the performance gain when a certain segment is stitched into the architecture.

## 3.1 Computation Graph Segmentation via Tokenization

We partition a CG $g$ into a contiguous sequence of subgraphs, which we call segments. A segment, denoted by $s$, may have multiple input and output nodes, or could even be a single primitive operation. For $g$ to be a valid neural network, any two contiguous segments $s_i$ and $s_j$ in $g$ must maintain the correctly matched height, width and channel $(H, W, C)$ resolutions. The resolutions of output nodes of $s_i$, have to match the resolutions of input nodes of its succeeding segment $s_j$. Our definition of segment spans a wider range of topologies and provides flexible operation grouping than predefined or handcrafted blocks, e.g., ResNet and MBConv blocks or DARTS cells [39].

The Segment Database $\mathcal{D}$ is the core component that stores the segment units that AutoGO mutations are based on. AutoGO uses the segment database $\mathcal{D}$ (or vocabulary) to either partition an input CG into segment units or select a segment from $\mathcal{D}$ to replace an existing segment in input CG. To alleviate the memory and time complexities of mining common subgraphs from a large number of CGs, we relax the problem into mining segments from sequences. This allows us to utilize much more efficient tokenization techniques over sequences.

Given a set of neural networks represented in a CG format $\mathcal{G} = \{g_1, \cdots, g_k\}$, we convert each CG into a topologically sorted sequence of nodes. We enrich node representation by labeling a node in the form of *[current op, incoming ops, outgoing ops]*. [3] Each unique node label is further encoded into a single character symbol. Thus, a topologically sorted CG can be mapped into a string of character symbols, where each character encodes an operation and its neighboring operations. By converting all graphs in $\mathcal{G}$ into sequences, we essentially have built a corpus for training a tokenizer to extract common segments of character symbols. Specifically, we use Byte-Pair Encoding (BPE), a data compression [20] technique with a wide use for text tokenization in NLP [60], to tokenize the string representations of CGs. BPE operates iteratively by collecting frequent pairs of consecutive symbols to build a vocabulary of tokens (segments). Using BPE, we extract the most common subsequences from the string representation of the CGs and build a vocabulary of size $|V|$. We revert each discovered subsequence in the vocabulary back to its corresponding subgraph representation from the CG to form a segment database $\mathcal{D}$. Given a new CG, BPE utilizes its built vocabulary and applies a greedy algorithm to segment it. Figure 1 provides an example of a segmented CG. Our approach brings several benefits over mining on large graphs with WL-kernels. The extraction process on sequences is efficient. Using BPE enables segment extraction from all benchmark families simultaneously without facing memory inefficiencies like WL-Kernel.

---

[3]For example, we label a Conv 3x3 node with incoming edges from Add and BatchNorm operations and an outgoing edge to a ReLU operation as "conv2d3,in,add,batchnorm,out,relu".

## 3.2 Computation Graph Mutation

We use an evolutionary search strategy to perform segment mutations on the CG of an input architecture and iteratively update a Pareto front of architectures in terms of predicted accuracy and a chosen hardware-friendliness objective, e.g., FLOPs, latency, and power. The mutation made to a parent architecture comprises the following steps: segmentation, source segment selection, replacement segment selection, tensor resolution (shape) propagation, and performance evaluation. First, AutoGO partitions the parent architecture into segments using the BPE-generated vocabulary $V$. BPE adopts a greedy segmentation approach, which will lead to deterministic partitioning. To diversify the segmentation outcome, we select a subset of vocabulary $V' \subset V$ that BPE uses during segmentation, thus leading to different partitioning outcomes every time the CG is segmented. After segmentation, we select a set of candidate source segments to mutate. For each source segment $s_i$, we randomly select multiple replacement segments $s_i^* \in \mathcal{D}$ that have the same number of inputs and outputs as the corresponding source segment. If $s_i^*$ has multiple inputs and/or outputs, AutoGO randomly maps these to the outputs and/or inputs created by removing $s_i$.

The mutation process must maintain a valid architecture, by correctly combining the replacement segment with the rest of the model. Given a CG $g$, let $\mathcal{S}_g = \{s_0, s_1, ..., s_{n-1}\}$ be the set of disjoint segment subgraphs generated by applying BPE to $g$. Let $s_i, 0 \le i \le n - 1$ be a source segment within $g$ that we wish to replace. We partition the segments within $\mathcal{S}_g$ into three distinct groups that reflect their positions within $g$:

- The **P**redecessor group denotes all segments $P = \{s_p; 0 \le p < i\}$ between the input and $s_i$.

- The specific **S**egment $s_i \in \mathcal{D}$ that we are aiming to replace by mutating $g$.

- The su**C**essor group denotes all the remaining network segments $C = \{s_c; i < c \le n - 1\}$.

Let $\{P, S, C\}$ refer to a CG partitioned in this manner, denoted by grey, purple and orange blocks in Fig. 3. Hence, for a mutation to be valid, the shape of the output tensor from the Predecessor $P$ must match that of the input to the replacement segment $s_i^*$ and the output shape of $s_i^*$ must match the shape of the expected input to suCcessor $C$. AutoGO adapts replacement segment $s_i^*$ to the remainder of the architecture $P$ and $C$ by adjusting the hyperparameters of operations in $s_i^*$ to achieve the desired resolutions. Adjustments are applied to mutable operations, e.g., increasing the stride of convolutions and pooling operations to induce downsampling, whereas operations such as activations and batch normalization are immutable. Depending on the $P$, $C$ and $s_i^*$ subgraphs, stitching the replacement segment into the overall CG may be infeasible. We cast this "resolution propagation" task as a Mixed-Integer Linear Program (MILP) over the

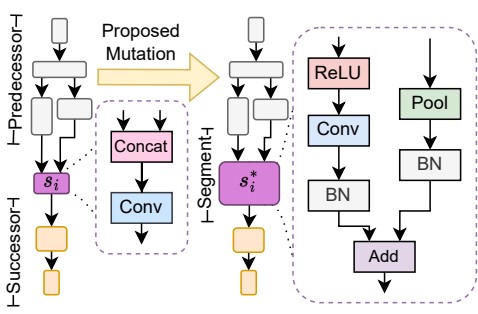

Figure 3: An example of segment mutation. AutoGO takes an input CG (left) and replaces a source segment $s_i$ with $s_i^*$. The predecessor (grey) and successor (yellow) are unchanged.

adjustable hyperparameters of mutable nodes within $s_i^*$, which is an optimization problem with linear objectives and constraints, and integer-valued decision variables. We define MILP constraints that regulate the correct resolution propagation within $s_i^*$ when stitched to the rest of the architecture.

At the end of each mutation iteration, we retain all the segment replacements that have a feasible solution to the resolution propagation MILP, and profile these valid segment mutations in terms of the predicted accuracy gain given by the PSC predictor (Sec. 3.3) and a selected hardware friendliness metric, based on which a Pareto front $\mathcal{O}$ [21] of architectures is maintained and updated. During the first iteration, we only mutate the input architecture. At the beginning of each successive iteration, AutoGO selects the $k$ architectures from the Pareto front as parents to mutate. If the Pareto front contains less than $k$ architectures, AutoGO will select additional non-Pareto optimal architectures having the minimum sum of accuracy and FLOPs ranks. Further technical details on the $\{P, S, C\}$ partitioning scheme, resolution propagation, parent selection process and overall algorithm, including examples, illustrations and pseudocode, are provided in supplementary Sections A.3.1 and A.5.

### 3.3 Context-Aware Mutation Performance Estimation

AutoGO uses a neural predictor to estimate the performance of mutant architectures in order to search for high-quality networks. We propose a novel predictor that assesses the potential performance benefit from a segment mutation, based on its topology, context and its location within the architecture.

Just as we can construct subgraphs from the individual segments produced by BPE, we can construct larger predecessor and successor subgraphs from all segments within $P$ and $C$, respectively. We use this format to train a **PSC** neural predictor. Let $h_*$ denote a fixed-length graph embedding for an arbitrary graph produced by a graph neural network (GNN) [51]. The PSC predictor operates by separately encoding the predecessor, segment, and successor CGs into distinct graph embeddings. These embeddings are then concatenated and fed into a multi-layer perceptron (MLP) regressor to predict the performance $y$ of $g$. Stated more formally:

$$h_P = \text{GNN}(P); \; h_{s_i} = \text{GNN}(s_i); \; h_C = \text{GNN}(C); \; y = \text{MLP}(\text{Concat}[h_P, h_{s_i}, h_C]).$$

It is also possible to predict performance using the entire CG, $y = \text{MLP}(\text{GNN}(g))$. However, our PSC predictor enjoys several advantages over this approach. By separately encoding the $\{P, S, C\}$ partitions, the PSC predictor is sensitive to the position and context of the segment $s_i$ within the overall network. This allows us to more directly estimate the performance impact that mutating $s_i$ will have. As shown in Figure 3, we are considering mutating $g$ into $g^*$ by replacing the source segment $s_i$ with a segment $s_i^*$. We want the mutant network to outperform the original, i.e., $y^* > y$. Our training process emphasizes learning a separate embedding for each $s_i$ in our segment database $\mathcal{D}$. It encodes the required knowledge to estimate the effect of small changes from segment $s_i$ to any replacement segment $s_i^*$, given a fixed $P$ and $C$.

## 4 Experimental Results

We construct our database by extracting segments from 5 CIFAR-10 [33] benchmark families: NAS-Bench-101 (NB-101) [71], NAS-Bench-201 (NB-201) [17], HiAML, Inception, and Two-Path [48]. Initially, we set the BPE vocabulary size to 2000 and obtain a database with 428 unique segments after filtering out isomorphisms. Segments vary in size ranging from primitives (containing a single operation node) to blocks with up to 16 nodes and edges. The average segment contains 5 nodes and 3 edges, and some segments are disconnected subgraphs spanning parallel branches of a CG. We provide in-depth statistics and visualizations in Section A.1.

In the remainder of this section, we apply AutoGO and our Segment Database to search for better architectures on several NAS benchmarking families to demonstrate the benefits of our framework. We further use AutoGO to improve several open-sourced, popular network architectures. We consider various high-resolution CV tasks, including Classification, Semantic Segmentation and Human Pose Estimation, with the aim of improving hardware friendliness in terms of FLOPs. We also provide examples of deployment where AutoGO minimizes the energy consumption or on-chip latency of already lightweight neural networks for Super Resolution and Image Denoising using a cycle simulator for a Huawei Neural Processing Unit (NPU). We provide implementation details, dataset metrics, and training setup in Sections A.2 and A.4.

### 4.1 Rank Correlation of Performance Predictors

AutoGO uses a neural predictor to estimate the performance of mutated architectures. We train and evaluate our PSC predictor on five CIFAR-10 benchmarks. Our CG format provides the advantage of training simultaneously on multiple benchmark architecture families. We split each family into training, validation, and testing partitions containing 80%, 10% and 10% of the overall CGs in that family. We combine the training

Table 1: Test SRCC of all 5 architecture families for the **PSC** predictor and baselines. Results averaged over 5 runs.

| Arch. Family | GNN | PSC 1:1 Ratio | PSC |
|---|---|---|---|
| NB-101 | $0.627 \pm 0.031$ | $0.666 \pm 0.025$ | $\mathbf{0.849} \pm 0.054$ |
| NB-201 | $0.809 \pm 0.016$ | $0.865 \pm 0.015$ | $\mathbf{0.983} \pm 0.003$ |
| HiAML | $0.010 \pm 0.013$ | $0.170 \pm 0.042$ | $\mathbf{0.734} \pm 0.031$ |
| Inception | $0.209 \pm 0.037$ | $0.066 \pm 0.071$ | $\mathbf{0.496} \pm 0.022$ |
| Two-Path | $0.023 \pm 0.018$ | $0.236 \pm 0.043$ | $\mathbf{0.724} \pm 0.022$ |

Table 2: AutoGO results across all 5 CIFAR-10 architecture families while aiming to increase accuracy [%] and reduce FLOPs [1e6]. We consider 3 experimental configurations that vary in unit of mutation and predictor used. We bold and italicize the best and second best result per family.

| | Baseline Architectures | | Operator + GNN | | Segment + GNN | | Segment + PSC | |
|---|---|---|---|---|---|---|---|---|
| Family | Acc. | FLOPs | Acc. | FLOPs | Acc. | FLOPs | Acc. | FLOPs |
| NB-101 | 95.18% | 11722 | 95.16% | 9407 | *95.31%* | *10817* | **95.45%** | **11118** |
| Δ | – | – | -0.02% | -19.75% | *+0.13%* | *-7.72%* | **+0.27%** | **-5.15%** |
| NB-201 | 93.50% | 313 | 93.37% | 232 | *93.57%* | *294* | **93.84%** | **303** |
| Δ | – | – | -0.13% | -25.88% | *+0.07%* | *-6.07%* | **+0.34%** | **-3.19%** |
| HiAML | 92.32% | 246 | 92.00% | 198 | *92.62%* | *168* | **92.75%** | **198** |
| Δ | – | – | -0.32% | -19.51% | *+0.30%* | *-31.71%* | **+0.43%** | **-19.51%** |
| Inception | 93.20% | 494 | 92.97% | 319 | *93.31%* | *461* | **93.52%** | **474** |
| Δ | – | – | -0.23% | -35.43% | *+0.11%* | *-6.68%* | **+0.32%** | **-4.05%** |
| Two-Path | 87.90% | 116 | 88.63% | 106 | **89.16%** | **48** | *88.94%* | *91* |
| Δ | – | – | +0.73% | -8.62% | **+1.26%** | **-58.62%** | *+1.04%* | *-21.55%* |

partitions for each family to form an overall training set for the predictors while setting the test partitions aside individually. When training the PSC predictor, we partition each CG into multiple $\{P, S, C\}$ instances to use as training samples. We compare our proposed **PSC** predictor with two baselines: As each CG contains multiple $\{P, S, C\}$ instances, we consider an intermediate setting, **PSC 1:1 Ratio**, where we only consider one random $\{P, S, C\}$ instance per CG in the training set. Moreover, we also consider a baseline **GNN** that estimates the performance of whole unpartitioned CGs but is not sensitive to segment-level changes.

We measure the Spearman's Rank Correlation Coefficient (SRCC) of each predictor on the test partitions for each benchmark family. SRCC is defined in the range [-1, 1] and higher values are better. Table 1 summarizes our results. We note the exceptional performance of the PSC predictor as it can obtain SRCC above 0.72 on HiAML and Two-Path while the GNN barely achieves positive SRCC. Moreover, while the GNN and PSC 1:1 predictors can obtain SRCC above 0.8 and 0.6 for NB-201 and NB-101, respectively, if we train the PSC predictors on all $\{P, S, C\}$ samples, we can obtain a near perfect SRCC of 0.98 on NB-201 and almost 0.85 on NB-101. Overall, these findings demonstrate the merit of our segment decomposition and PSC encoding scheme for CGs.

## 4.2 Improving CIFAR-10 NAS Benchmark Architectures

We test the effectiveness of AutoGO by refining the best architectures from each family. Specifically, AutoGO aims to increase accuracy while reducing FLOPs. We consider three scenarios that allow us to ablate the effectiveness of our PSC predictor when applied to search. We also compare our segment-level mutation to a simpler, operation-level mutation that mutates single primitive operation.

We run AutoGO for 10 iterations in the segment-level mutation, and 50 iterations for the operation-level mutation for a fair comparison since segments have 5 nodes on average. At the end of each iteration, we randomly select 10 architectures from the accuracy-FLOPs Pareto frontier to qualify for the next iteration as parent architectures. For each parent candidate, we consider up to 100 replacement mutations. We allow AutoGO to mutate sequences of 1 to 3 contiguous source segments simultaneously. We randomly mask out 50% of segments with more than 1 node in our segment database and force BPE to segment the input with the remaining ones. Further, we consider two search settings that limit the FLOPs decrease of child architectures. In the first case, we allow AutoGO to freely reduce FLOPs, while in the second case, we do not allow FLOPs to fall by more than 20% relative to the original network we are optimizing. After the search is complete, we train architectures on the accuracy-FLOPs Pareto frontier 3 times on CIFAR-10 [33]. We report the accuracy and FLOPs of the overall best architecture found across both FLOPs constraint settings. Finally, it takes 45 to 90 minutes to execute the search depending on the size of the input architecture CG. We provide an ablation study across FLOPs constraints, a detailed breakdown of wall-clock time cost, and enumerate our hardware platform in Sections A.6, A.7 and A.9, respectively.

Table 2 reports our findings across all 5 architectures families. We observe that the segment-level mutation is a better fit for finding high-performance architectures, as the best architectures are found using it. For example, on HiAML, it can increase the accuracy by up by 0.43% while reducing

Table 3: Results running AutoGO on Computation Graphs for ResNet-50, 101 and VGG-16. Specifically, we compare ImageNet [58] Top-1/Top-5 accuracy, Cityscapes test mIoU [14] using a PSPNet head [76], MPII [4] PCK as well as FLOPs. For performance metrics, higher is better. We measure latency on an Nvidia RTX 2080 Ti GPU using an input resolution size of $224 \times 224$.

| Architecture | ImageNet Top-1/5 | Cityscapes mIoU | MPII PCK | FLOPs [1e9] | Lat. [ms] |
|---|---|---|---|---|---|
| ResNet-50 Original | 74.02%/91.22% | 63.42% | 82.36% | 6.29 | 7.18 |
| ResNet-50 AutoGO Arch 1 | 75.34%/92.16% | 65.88% | **84.07**% | 6.71 | 7.50 |
| ResNet-50 AutoGO Arch 2 | **75.66**%/**92.45**% | **66.65**% | 82.70% | 5.88 | 6.92 |
| ResNet-101 Original | 75.09%/91.94% | 65.92% | 82.77% | 13.76 | 15.86 |
| ResNet-101 AutoGO Arch 1 | **76.56**%/**93.09**% | **67.12**% | 83.59% | 13.66 | 15.56 |
| ResNet-101 AutoGO Arch 2 | 75.69%/92.15% | 66.38% | **84.64**% | 13.35 | 15.36 |
| VGG-16 Original | 74.18%/91.83% | 65.36% | 85.92% | 30.81 | 4.65 |
| VGG-16 AutoGO | **74.91**%/**93.23**% | **66.91**% | **85.99**% | 24.34 | 4.20 |

FLOPs by -19.76%. By contrast, the operation-level mutation only manages to improve performance on Two-Path. However this gain of +0.73% is substantially less than what we can achieve using segment mutation. On NAS-Bench-101, operation mutation manages to break even with the baseline architecture, while incurring an accuracy drop of more than 0.10% on all other families.

Next, we compare the effectiveness of the PSC and GNN predictors. The PSC predictor finds the best architecture in 4 of the 5 architecture families. PSC improves accuracy on NB-101 and NB-201 by 0.34% and 0.27%, respectively. By contrast, the GNN only achieves the best performance on Two-Path, which is the smallest benchmark family with a baseline architecture of only 116 MegaFLOPs. Thus, segment-level mutations span more considerable changes that are distinguishable with GNN and PSC predictors. In sum, the segment-aware encoding is better at increasing performance while reducing FLOPs than the operation-level mutation. Moreover, our results demonstrate the superiority of the PSC predictor compared to the GNN in most cases.

### 4.3 Application to High-Resolution Classification, Segmentation and Pose Estimation

To demonstrate the extensibility and generalizability of our framework, we apply it to several stand-alone architectures for higher-resolution computer vision tasks. Specifically, we perform NAS using AutoGO with the PSC predictor and segment-level mutation for 5 iterations on ResNet-50, ResNet-101 [24] and VGG-16 [61]. For a fair comparison, we do not allow AutoGO to select segments containing operations that were not available or popularized when the network was first proposed, e.g., depthwise convolutions [59].

After the search, we examine the architectures on the Pareto frontier and select 1-2 with noticeably different FLOPs reductions to train and evaluate against the original

Figure 4: Example of a segment mutation from that helped create ResNet-50 AutoGO Arch 2 (Tab. 3). A ResNet block is replaced by a HiAML block.

architecture. To form the first point of comparison, we train each network on ImageNet [58]. Then, we fine-tune the network on different tasks. For Semantic Segmentation (SS), we use a PSPNet [76] head structure and fine-tune on Cityscapes [14] to obtain mean Intersection over Union (mIoU) performance. For Human Pose Estimation (HPE), we adopt the method of [78] to fine-tune on MPII [4] to measure the Percentage of Correct Keypoints (PCK) of an architecture.

Table 3 shows our results on ImageNet, Cityscapes, and MPII. First, we note how in every case, the architectures generated by AutoGO consistently outperform the original on all 3 CV benchmarks.

Table 4: Peak Signal-to-Noise Ratio (PSNR) for EDSR on the DIV2K validation set and several SR benchmarks in the 2x upscaling setting. Higher is better. We measure latency on an RTX 2080 Ti.

| SR Architecture | DIV2K | Set5 | Set14 | BSD100 | Urban100 | Manga109 | FLOPs [1e9] | Lat. [ms] |
|---|---|---|---|---|---|---|---|---|
| EDSR Original | 36.19 | 36.86 | 32.57 | 31.39 | 29.14 | 36.09 | 141 | 18.04 |
| EDSR AutoGO Arch 1 | **37.28** | **38.01** | **33.62** | **32.18** | **31.56** | **38.49** | 118 | 15.38 |
| EDSR AutoGO Arch 2 | 37.27 | 37.97 | 33.55 | 32.16 | 31.53 | 38.47 | 110 | 14.52 |
| EDSR AutoGO Arch 3 | 37.25 | **38.01** | 33.58 | 32.16 | 31.46 | 38.44 | 105 | 13.81 |

Table 5: SR PSNR results on Proprietary FSRCNN networks. FSRCNN-{3, 4} denotes the number of Conv3x3 operations in the middle of the architecture. We report change in power according to a cycle-accurate simulation model that uses a 64x640 input resolution. FLOPs [1e9] are measured using an input resolution of 640x360.

| SR Architecture | Set5 | Set14 | BSD100 | Urban100 | Manga109 | Power [mW] | ΔPower | FLOPs |
|---|---|---|---|---|---|---|---|---|
| FSRCNN-3 Original | **35.12** | **31.43** | **30.56** | **27.65** | **32.75** | 774.77 | – | 2.67 |
| FSRCNN-3 AutoGO | **35.12** | **31.43** | **30.56** | 27.64 | 32.60 | 644.10 | -16.87% | 2.09 |
| FSRCNN-4 Original | **35.22** | 31.50 | **30.60** | **27.71** | **32.88** | 892.89 | – | 3.74 |
| FSRCNN-4 AutoGO | 35.17 | **31.52** | **30.60** | **27.71** | 32.77 | 508.37 | -43.06% | 2.07 |

For example, ResNet-50 AutoGO Arch 2 outperforms the original by over 1.64% ImageNet top-1 accuracy, while the found architecture on VGG outperforms the original on Cityscapes by 1.55% mIoU. Also, we measure inference latency on an RTX 2080 Ti GPU. We note some correlation between FLOPs and GPU latency; as one metric increases or decreases, so does the other metric.

Figure 4 illustrates how AutoGO splices a HiAML segment into ResNet-50 to create a new architecture. The longer branch performs multiple convolutions at reduced channels, while the shorter branch applies lightweight operations on the original number of channels, and the MILP performs resolution propagation to ensure functionality.

### 4.4 Application to Super Resolution with EDSR

We use AutoGO to optimize networks for Super Resolution (SR). Specifically, we optimize the backbone feature extractors of EDSR [38]. As the original EDSR only uses convolution and ReLU operations, we do not let AutoGO select segments that contain depthwise, pooling, or batch normalization. Figure 9 (Sec. A.8) provides sample illustrations of the mutations AutoGO performs on EDSR. We train SR networks on DIV2K [2, 29] in the 2x resolution setting and evaluate on several public benchmarks [8, 74, 44, 28, 45, 3]. Table 4 demonstrates how EDSR architectures produced by AutoGO can handily outperform the original network while substantially reducing FLOPs and GPU latency, e.g., AutoGO Arch 3 is 36 gigaFLOPs smaller and 4.2ms faster.

### 4.5 Using AutoGO to Automate Neural Network Deployment on a Neural Processing Unit

Table 6: Results of using AutoGO to optimize a Proprietary U-Net Denoising network to improve PSNR and minimize on-chip latency. We report changes in latency and power measured on a mobile NPU using a cycle simulation model.

| Denoising | PSNR | ΔLatency | Power [mW] | ΔPower | FLOPs [1e9] |
|---|---|---|---|---|---|
| Base Model | 139.4 | – | 724.59 | – | 17.05 |
| AutoGO | **139.9** | -24.94% | 657.82 | -9.21% | 16.26 |

We demonstrate the real-world deployment capabilities of AutoGO by optimizing neural network performance using a cycle-accurate counter that simulates Huawei NPU performance for cell-phones [37]. We optimize for power or on-chip latency by pairing our pretrained PSC accuracy predictor with power/latency measurements fed back by either a hardware profiling tool or the cycle-accurate hardware simulator.

**Super Resolution Power Optimization** We use AutoGO to optimize proprietary lightweight SR models similar to FSRCNN [16]. Table 5 reports the network performance on several public datasets as well as the change in power and FLOPs. We note the effectiveness of AutoGO at optimizing the energy efficiency of even a small network, e.g., the FSRCNN-4 AutoGO variant can reduce the

instantaneous power of an already small FSRCNN (with 4 Conv3x3 in the body network) by over 43%, while the FSRCNN-3 AutoGO variant reduces it by over 16%. Moreover, AutoGO maintains or even enhances the PSNR performance as compared to the original networks, demonstrating the generalizability of the pretrained PSC predictor to other tasks.

**Image Denoising Latency Optimization** We use AutoGO to optimize a proprietary Image Denoising U-Net similar to [55] to reduce on-chip latency. Table 6 reports our findings on an in-house dataset. We observe how the mutated network can exceed the original denoising PSNR by 0.5. While substantially improving latency, we have also reduced other resource consumption metrics including power and FLOPs.

## 5 Limitations and Future Discussions

The AutoGO framework consists of many components: frequent subgraph mining (FSM) via topological sorting and BPE tokenization, the position-aware PSC predictor, the mutation-based evolutionary strategy and use of Computational Graphs. Each of these components has its own strengths and weaknesses. Our paper demonstrates the feasibility of using topological sorting and BPE to perform FSM, although it faces limitations due to the non-deterministic nature of topological sorting, resulting in generation of segments for isomorphic subgraphs that must be filtered out from our database. However, the primary strength of FSM through topological sorting and BPE is the economic advantage of speed, as neither BPE-based segment extraction nor isomorphic segment filtering is time-consuming, even on large Computational Graphs. Moreover, FSM only considers the frequency of a given subgraph (represented as a segment) while ignoring its contribution to performance and hardware-friendliness metrics. Extracting frequent subgraphs that can explain performance is a subject for further studies.

The quality of our segment database depends on the types of operations and subgraphs present in the benchmark families we extract from. For example, a few segments in our database use depthwise convolutions (see Fig. 5 in Sec. A.1) as the only NAS-Benchmark we consider that contains them is Inception, and in limited quantity. These were not widely used in our experiments, since to achieve a fair comparison with the baseline architectures that AutoGO aimed to improve, we constrained the vocabulary of AutoGO during the search to use only same generation of operations. On the flip side, one could use AutoGO to mine newer operations like depthwise convolutions, StarReLU [72], or even older operations that have gained popularity like GELU [25]. Segments containing these operations could then be used to further refine older architectures like ResNets, EDSR, and FSRCNN for performance improvement.

A future variant of AutoGO could replace the mutation-driven search with a policy network [35] to select replacement segments. Another avenue for future research is performing frequent and important computational subgraph mining for Transformer and attention-based models for their hardware-friendly deployment, as the Computational Graph representation and subgraph mining presented in this paper are principally designed for convolutional neural networks currently.

## 6 Conclusion

We propose AutoGO, or Automatic Graph Optimization, a new framework for optimizing neural networks outside the bounds of predefined, fixed search spaces. AutoGO represents architectures using a computation graph format of primitive operations. We partition computation graphs into segment subgraphs using Byte-Pair Encoding. Using a segment database and guided by a predictor which is sensitive to segment size and position, AutoGO modifies the network by incrementally mutating its segments while a resolution propagation MILP ensures network functionality. We build a segment database by extracting a vocabulary of segments from 5 open-source NAS benchmarks using Frequent Subgraph Mining. We use AutoGO to improve the accuracy of the best architectures from each of the 5 CIFAR-10 search spaces while reducing FLOPs. Furthermore, we use AutoGO to evolve several open-sourced large CNNs, including ResNets, VGG-16, and EDSR, and successfully improve their performance on a breadth of CV tasks with reduced or comparable FLOPs. Finally, we demonstrate how to utilize AutoGO to automatically reduce the hardware energy consumption and on-chip latency of realistic convolutional neural network applications, when deployed onto a mobile Neural Processing Unit.

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

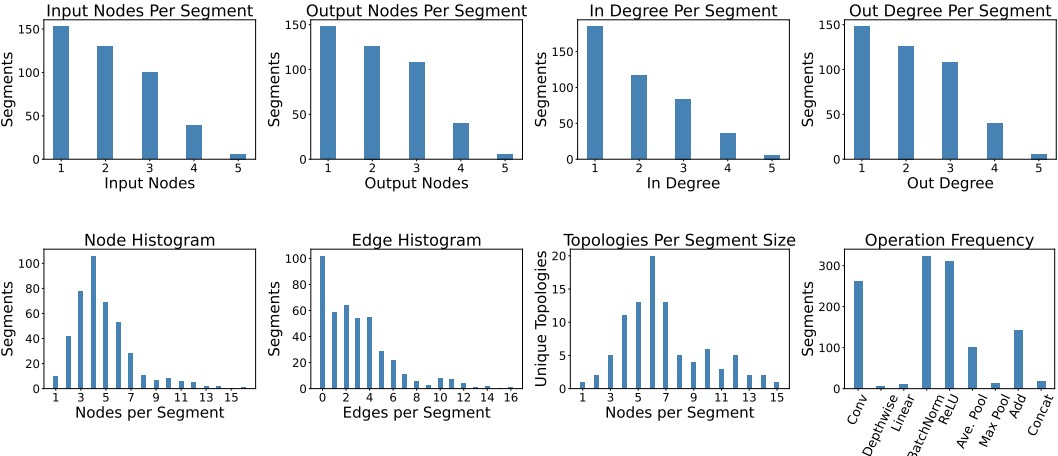

Figure 5: Histograms of segment database statistics including number of input and output nodes/degrees, nodes/edges per segment, unique segment topologies and operation frequency.

# A    Supplementary Material

## A.1    Additional Database Statistics

Figure 5 provides histograms regarding our segment database. Additionally, we enumerate the primitive operations that are only present in specific NAS-Benchmark families:

- **Depthwise**: Inception.
- **Max Pool**: NB-101, Inception and Two-Path.
- **Concat**: NB-101, Inception and Two-Path.

All other operation primitives, e.g., Conv, ReLU, BatchNorm, etc., are present across all 5 CIFAR-10 NAS-Benchmarks.

## A.2    Predictor and Dataset Details

We further elaborate on the baseline GNN and PSC predictors from Section 4.1. We provide implementation details, dataset statistics and data pre-processing techniques. We train our predictors for 40 epochs with a batch size of 32 and an initial learning rate of $1e^{-4}$.

### A.2.1    Baseline and PSC Predictor Setup

We use the same baseline GNN predictor as GENNAPE [48]. First, CGs are given as input into an initial set of embedding layers that transform discrete node features, such as operation type, input/output tensor resolution, kernel size, and bias, into a continuous vector. The node embeddings are then fed through a series of 6 $k$-GNN [51] layers. Next, an overall graph embedding is computed by taking the mean of all node embeddings. A simple MLP with 4 hidden layers predicts performance using the graph embedding.

The PSC predictor differs in that each CG sample is first split into its respective *Predecessor*, *Segment*, and *suCcessor* subgraphs before being fed into the predictor. All three subgraphs are processed as separate CGs by the node embedding and $k$-GNN layers to produce three distinct graph embeddings. We concatenate these graph embeddings feature-wise and feed them into an MLP to generate a prediction. Also, node embedding and $k$-GNN layer weights are shared for each subgraph type.

### A.2.2 Dataset Statistics and PSC Preprocessing

Table 7: Number of Computation Graphs (CG), segment samples and test SRCC folds for each family. We randomly sample 5k NB-101 architectures and only consider NB-201 networks that do not have the 'none' operation.

| Arch. Family | CGs | Segments | Folds |
| --- | --- | --- | --- |
| NB-101 | 5.0k | 404.9k | 42 |
| NB-201 | 4096 | 252.8k | 34 |
| HiAML | 4.6k | 65.1k | 10 |
| Inception | 580 | 222.4k | 129 |
| Two-Path | 6.9k | 193.1k | 10 |

We train and evaluate the baseline GNN predictor on every unique CG sample. Additional steps are required to train the PSC predictor since each CG comprises many segments and can decompose into many distinct $\{P, S, C\}$ subgraph sets.

For the intermediate baseline, **PSC 1:1 Ratio** in Table 1, we randomly sample 1 $\{P, S, C\}$ representation from each segmented CG in our training dataset. Hence, the number of samples equals the original number of training instances. For the full **PSC** predictor, we remove this restriction and consider all possible $\{P, S, C\}$ decompositions which drastically increases the number of samples.

Table 7 lists the number of CGs and $\{P, S, C\}$ samples per family. While each $\{P, S, C\}$ sample for a given CG focuses on a different network segment, they still describe the same overall architecture and thus retain the same accuracy label. Therefore, when measuring test SRCC on the PSC predictor, we divide the test data into *folds*. Each fold contains only one $\{P, S, C\}$ instance of a given CG. This avoids introducing additional ties in the ground-truth labels when calculating SRCC. The number of folds is equal to the minimum number of segments in any test CGs or 10, whichever is smaller. Therefore, we calculate the overall test SRCC by averaging SRCC across each fold.

### A.3 Segment Extraction with BPE

We compare our BPE subgraph extraction approach to the Weisfeiler-Leman (WL) Kernel method adopted by NAS-BOWL [56] in terms of efficiency. NAS-BOWL applied it on the original, shallow cell-based network representation of NAS-Bench-201 with a depth of 2. We use the WL-kernel on the CG-level and enumerate all subgraphs with a maximum depth of 5. The time and RAM costs of using the WL-kernel scale poorly as we increase the number of graphs and nodes per graph. For example, it takes at least 6 hours to extract and count subgraphs from each NAS-Benchmark family. Moreover, we could not use more than 1k CGs from the HiAML or NB-201 families (~110 and ~250 nodes per CG, respectively) without facing memory issues on the rack server described in Section A.9.

By contrast, our approach brings several benefits over mining on large graphs with WL-kernels. The extraction process on sequences is efficient. Using BPE enables segment extraction from all benchmark families (over 21k CGs per Tab. 7) simultaneously in less than 20 minutes using around 10GB of RAM. Also, BPE provides segments that are easier to mutate and alleviates limitations with WL-kernel extraction process by topologically ordering the nodes. Figure 6 compares WL and BPE segmentations on a part of a CG from the NAS-Bench-201 family. The subgraph extracted from the WL method (Fig. 6(a)) cannot cover several nodes within its context (grey nodes of BN-8, Pool-10, BN-11, and Add-14) due to a limited depth of 5 and several resid-

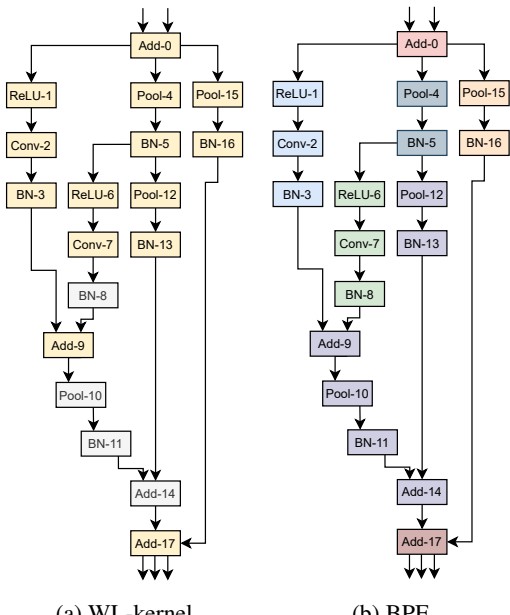

(a) WL-kernel  (b) BPE

Figure 6: Comparison between subgraphs extracted with WL-kernel and BPE on a NAS-Bench-201 cell. Nodes are numerically labeled by a topological ordering. Best viewed in color. Specifically, WL-kernel extracts one large subgraph consisting of all highlighted nodes (greyed-out nodes are omitted). For BPE, all nodes are extracted into one subgraph, denoted by a unique color.

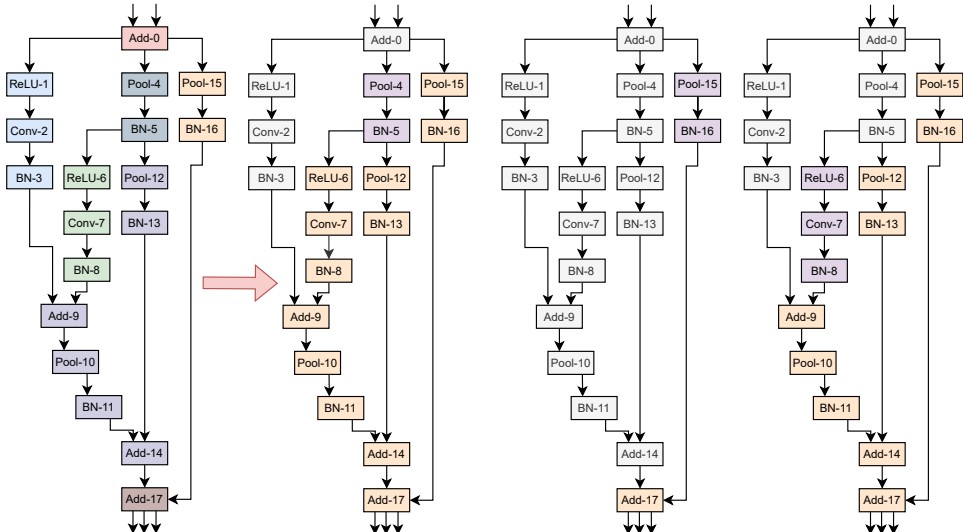

Figure 7: Example of how the BPE-segmented graph in Figure 6(b) is partitioned into Predecessor, Segment and suCcessor subgraphs based on the selected segment. Specifically, we highlight nodes of the selected segment in purple, the predecessor in grey and the successor in yellow.

ual connections. This exacerbates the mutation process. In contrast, segmentation with BPE (Fig. 6(b)) spans different subgraph sizes denoted by separate colors.

### A.3.1 PSC Partitioning with Parallel Branches

Figure 7 illustrates how a Computational Graph with multiple branches can be cleanly partitioned into different $\{P, S, C\}$ based on the choice of segment $s_i$. Although the input CG has multiple parallel branches, the use of topological sort to assign ordered numerical labels to each node. The numerical labels of each node in a segment are contiguous, while parallel branches are assigned to either the Predecessor or suCcessor according to their numerical labels.

## A.4 Architecture Training Hyperparameters

We elaborate on the training recipes we use to evaluate input baseline architectures as well as those found by AutoGO.

### A.4.1 CIFAR-10 Families

We use the CG representation of the initial and mutated architectures to instantiate networks and train them using TensorFlow. We evaluate CIFAR-10 networks by training them 3 times for 200 epochs with a batch size of 256. We optimize the models using RMSProp with an initial learning rate of $1e^{-3}$ and a momentum factor of 0.9. We anneal the learning rate according to a cosine schedule.

### A.4.2 ImageNet, Segmentation and Pose Estimation

When evaluating ResNet and VGG[4] architectures, we first train on ImageNet [58] using timm [66] with a batch size of 1024. We use an initial learning rate of 0.1 which we anneal using a cosine schedule. We optimize the model using Stochastic Gradient Descent (SGD) with a momentum factor of 0.9 and a weight decay of $1e^{-4}$. We set a gradient clipping value of 5.0 and use label smoothing with $\epsilon = 0.1$. We train ResNets for 200 epochs and VGG-16 for 100 epochs. We save the trained weights to fine-tune on other tasks.

We evaluate Semantic Segmentation performance using semseg [75]. The PSPNet [76] head requires two inputs to implement properly. The first is the final latent tensor that originally feeds into the

---

[4]Base model uses batch normalization

classifier head, while the second requires grafting an auxiliary residual connection $3/4$ths of the way through the network feature extractor. Furthermore, we adjust the dilation factor and strides of all convolution and pooling operations in the later part of the network to limit downsampling. After loading the pretrained ImageNet weights, we fine-tune on Cityscapes [14] images cropped to $713^2$ for 200 epochs using a batch size of 16. We use SDG with an initial learning rate of 0.01, a momentum factor of 0.9, and a weight decay of $1e^{-4}$.

We implement 2D Human Pose Estimation using [78]. To convert an ImageNet network, we remove the classifier layers and then append a series of 'Deconvolution-BatchNorm-ReLU' blocks which gradually upsample the latent tensors from $8^2$ to $64^2$. We train on MPII [4] images cropped to $256^2$ for 140 epochs with a batch size of 32. We optimize our networks using Adam, setting an initial learning rate of $1e^{-3}$ for ResNet-50 and VGG-16, and $5e^{-4}$ for ResNet-101. We reduce the learning rate by a factor of 10 at epochs 90 and 120. Finally, we report performance in terms of the Percentage of Correct Keypoints (PCK), specifically the Percentage of Correct Keypoints at a head-neck distance of 0.5 (PCK@h0.5) [77].

### A.4.3 Super Resolution

We train networks on DIV2K in the 2x upsampling setting for 1000 epochs with a batch size of 16. We set an input patch size of 64 for EDSR and 48 for FSRCNN. We minimize the L1 loss using the Adam optimizer with an initial learning rate of $1e^{-4}$, which we reduce using a cosine decay schedule.

### A.4.4 Image Denoising

We train networks on a custom in-house image-denoising dataset with 7k images. We set an input patch size of 128 for all networks. We train each network for 2k epochs under a batch size of 128. We minimize the L1 loss using the Adam optimizer with an initial learning rate of $1e^{-3}$ and a final learning rate of $1e^{-6}$, reduced over a polynomial schedule.

### A.5 Additional AutoGO Search Details

We provide additional details on the AutoGO search algorithm from Section 3.2.

---

**Algorithm 1** Sample AutoGO pseudocode for one iteration

---

1: **Input:** Pareto frontier $\mathcal{O}$             ▷ Only contains the input architecture at iteration 0.
2: **Input:** Segment Database $\mathcal{D}$
3: **Input:** Performance Predictor $p$ and FLOPs counter $f$.
4:   $G_k = Sample(\mathcal{O}, k)$                     ▷ Sample $k$ architectures
5: **for** $g \in G_k$ **do**
6:     $PSC_g = []$                      ▷ Empty list of mutants
7:     $\mathcal{S}_g = Segment(g, \mathcal{D})$
8:     **for** $s \in Sample(\mathcal{S}_g)$ **do**                ▷ Source segments
9:       $\{P, s, C\} = Partition(\mathcal{S}_g, s)$
10:       **for** $s^* \in Sample(s, \mathcal{D})$ **do**        ▷ Sample replacement segments
11:         $\{P, s^*, C\} = Mutate(P, s^*, C)$
12:         **if** $MILP(\{P, s^*, C\})$ finds a solution **then**    ▷ Resolution propagation
13:           Add $\{P, s^*, C\}$ to $PSC_g$
14:         **end if**
15:       **end for**
16:     **end for**
17:     **for** All mutated $\{P, s^*, C\} \in PSC_g$ **do**
18:       Profile $\{P, s^*, C\}$ using $p$ and $f$
19:       Update $\mathcal{O}$ using $\{P, s^*, C\}$ and its profiled information.
20:     **end for**
21: **end for**

---

### A.5.1 AutoGO Pseudocode Algorithm

Algorithm 1 provides an example of how the AutoGO search procedure executes over one iteration. AutoGO selects parent architectures from the Pareto frontier $\mathcal{O}$. It then uses the segment database $\mathcal{D}$ to select source and replacement segments to create mutant child architectures. The resolution propagation MILP ensures the mutants constitute valid architectures. Finally, AutoGO places the child architectures on the Pareto frontier $\mathcal{O}$ according to their performance and hardware-friendliness.

### A.5.2 Node Labeling

Before segmentation with BPE, we label nodes in the CG in the form of *[current operation, incoming operations, outgoing operations]*. We encode each unique node label with a single Chinese character symbol, as they span a wide range of symbols compared to other languages.

### A.5.3 Selecting a Sparse BPE Vocabulary

When generating $V'$ as a vocabulary set utilized by BPE to segment CGs, we include all single-node segments as these represent the irreducible primitive operations that must exist within the vocabulary in some form and only filter out multi-node segments.

### A.5.4 Selecting Non-Pareto Optimal Architectures

When transitioning from iteration $e$ to $e + 1$, we select $k$ architectures from the Pareto frontier $\mathcal{O}$ and search history to serve as parents. If we have sufficient architectures on the Pareto frontier, $|\mathcal{O}| \geq k$, we randomly sample from it. However, if $|\mathcal{O}| < k$, there is an architecture deficit. We compensate for this deficit by selecting non-Pareto optimal architectures that aim to achieve our search objective. We select these architectures by ranking them in terms of predicted accuracy and FLOPs, where higher and lower are better, respectively. We then sum these ranks and select the non-Pareto optimal architectures with the lowest rank sum. Table 8 provides a simple example of this process. Note how the selection mechanism excludes architectures that have high performance but are too large, as well as underperforming architectures.

Table 8: Example of the minimum sum of ranks selection algorithm with a deficit of 3 architectures.

| Acc. [%] | Rank | FLOPs | Rank | Rank Sum | Selected? |
|---|---|---|---|---|---|
| 91.21 | 0 | 260 | 5 | 5 | No |
| 91.10 | 1 | 215 | 2 | 3 | **Yes** |
| 91.02 | 2 | 200 | 0 | 2 | **Yes** |
| 90.75 | 3 | 210 | 1 | 4 | **Yes** |
| 90.35 | 4 | 220 | 3 | 7 | No |
| 89.05 | 5 | 250 | 4 | 9 | No |

### A.5.5 Segment Selection

For each CG $g$, we sample a set of $m$ source segments $s_i$. We sort the segments $\mathcal{S}_g$ by FLOPs and then we select the $m/2$ segments with the lowest FLOPs while randomly sampling the rest.

### A.5.6 Accuracy Predictions and FLOPs Constraints

Once we have a set of valid source and replacement segments, we use the PSC predictor to select mutations that yield the most significant accuracy gain. We use a FLOPs calculator (or a proprietary profiling tool for measuring NPU latency/power) to further filter these mutations by rejecting child architectures whose FLOPs deviate too far from the FLOPs of the input architecture.

### A.5.7 Resolution Propagation

Adjustment can not always lead to a solution, meaning the replacement segment can not be used for mutation at this position and generate a valid CG. We cast this task as a search problem over the height, width, and channel resolution values on the replacement segment operations. The search spans mutable operations such as convolutions and pooling. The rest of the operations are immutable and only forward the resolution without changing its sizes, such as add, activation functions, and batch normalization. During the search, we limit the adjustments on the values of height, width, and channel sizes to doubling, halving, or keeping the same.

Our solution is based on Mixed Integer Linear Programming (MILP). MILP is an optimization problem formulated with linear objectives, linear constraints, and integer-valued variables. The input to MILP is the replacement segment DAG. Each node has two variables per each height, width, and channel dimension, denoting input and output resolutions. Each edge is associated with a "flow" variable. We define MILP constraints that regulate the correct flow of resolution. Immutable nodes have input resolutions equal to output resolutions. The output resolution for mutable nodes is less than or equal to the input resolution. The model is optimized to achieve the expected resolution at the output nodes. The model is proven infeasible if the search fails to achieve expected output resolutions.

We briefly illustrate the resolution propagation process. Figure 8 shows a replacement segment (yellow) that is being put together with the Predecessor (blue) and Successor (green) partitions of the network. We provide the output resolution of each operation in the form of (height, width, channel). Notice how the number of input and output nodes of the replacement segment matches the number of output and input nodes of the Predecessor and Successor, respectively. Initially, the replacement segment expects input dimension sizes for its 'Conv' and 'BN' operations of (32, 32, 16), which are the resolutions of the Predecessor's output nodes. Also, the Successor expects an input size of (16, 16, 32), which demands the replacement segment to output a feature map with this dimension at the 'Add' operation. This requires adjusting the resolution of

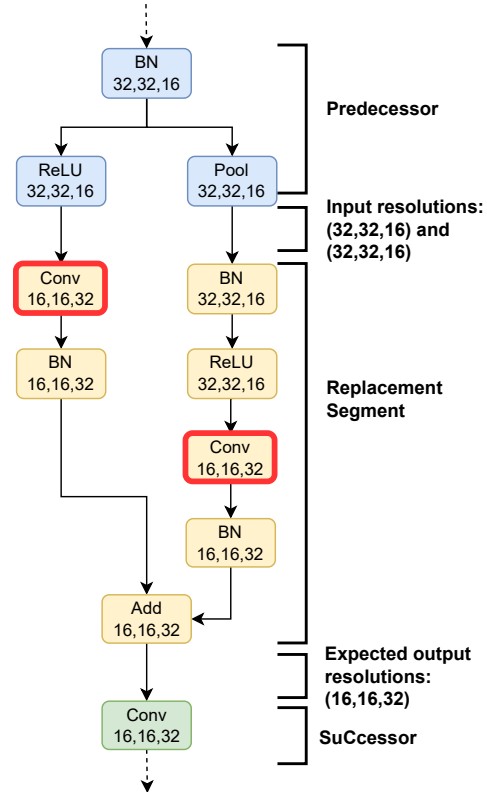

Figure 8: Resolution propagation adjusts the resolution of mutable operations in the replacement segment. The Height, Width, and Channel sizes are adjusted in both 'Conv' operations so that the replacement segment yields the expected output resolution at the 'Add' operation.

the 2 mutable 'Conv' operations in the replacement segment (highlighted with red borders). Notice that adjusting one of them or leaving resolutions unadjusted will result in incorrect propagation because the 'Add' operations require its incoming tensors to have the exact same dimensions. We use MILP to solve this problem by finding the correct adjustment to mutable operations by halving, doubling, or maintaining resolution sizes.

## A.6 CIFAR-10 FLOPs Restraint Ablation

Table 9 provides a full ablation study of AutoGO on all 5 CIFAR-10 families in terms of FLOPs reduction constraint. We consider two settings where AutoGO can reduce FLOPs by at most -20% relative to the baseline architecture, or can reduce them freely (-100%), while always limiting FLOPs *increases* to be at most +10%. We again note how the best architecture for each family was found using segment mutations.

We observe that the segment-level mutation is a better fit for finding high-performance architectures under wider FLOPs constraints. For example, on HiAML, the segment-mutation cannot improve the accuracy of the base architecture when we impose a FLOPs reduction limit of -20%, yet it can increase the accuracy by up to 0.43% on average when we remove the restriction, even though the best architecture only reduces FLOPs by -19.76%. From this result, we infer that FLOPs restrictions hamper the exploration of the segment-level mutation. The only family where the -20% FLOPs constraint produces a better architecture than the no-constraint setting is Inception, which is already the second-largest family with a base model size of nearly 500 MegaFLOPs. By contrast, the operation-level mutations require FLOPs reduction constraints to break even with the baseline

Table 9: Full ablation study of AutoGO on all 5 CIFAR-10 families considering choice of mutation unit {Operation, Segment}, predictor {GNN, PSC} and FLOPs [1e6] reduction ($\delta$) constraint {-20%, -100%}, extending the results of Table 2. For each experiment, we report the accuracy [%] and FLOPs [1e6] (raw and $\Delta$ relative to the baseline). We bold and italicize the best and second best result per family, respectively.

| Family ($\delta$FLOPs) | Baseline | | Operator + GNN | | Segment + GNN | | Segment + PSC | |
|---|---|---|---|---|---|---|---|---|
| | Acc. | FLOPs | Acc. | FLOPs | Acc. | FLOPs | Acc. | FLOPs |
| NB-101 (-20%) | 95.18% | 11722 | 95.16% | 9407 | *95.31%* | *10817* | 95.06% | 9606 |
| $\Delta$ | | | -0.02% | -19.75% | *+0.13%* | *-7.72%* | -0.12% | -18.05% |
| NB-101 (-100%) | | | 93.12% | 1591 | 95.25% | 10513 | **95.45%** | **11118** |
| $\Delta$ | | | -2.06% | -86.43% | +0.07% | -10.31% | **+0.27%** | **-5.15%** |
| NB-201 (-20%) | 93.50% | 313 | 93.28% | 250 | 92.86% | 250 | 93.32% | 251 |
| $\Delta$ | | | -0.22% | -20.13% | -0.34% | -20.13% | -0.18% | -19.81% |
| NB-201 (-100%) | | | 93.37% | 232 | *93.57%* | *294* | **93.84%** | **303** |
| $\Delta$ | | | -0.13% | -25.88% | *+0.07%* | *-6.07%* | **+0.34%** | **-3.19%** |
| HiAML (-20%) | 92.32% | 246 | 92.00% | 198 | 92.08% | 198 | 92.22% | 230 |
| $\Delta$ | | | -0.32% | -19.51% | -0.24% | -19.51% | -0.10% | -6.50% |
| HiAML (-100%) | | | 84.63% | 28 | *92.62%* | *168* | **92.75%** | **198** |
| $\Delta$ | | | -7.69% | -88.62% | *+0.30%* | *-31.71%* | **+0.43%** | **-19.51%** |
| Inception (-20%) | 93.50% | 494 | 92.97% | 399 | 93.12% | 399 | **93.52%** | **474** |
| $\Delta$ | | | -0.23% | -19.23% | -0.08% | -19.23% | **+0.32%** | **-4.05%** |
| Inception (-100%) | | | 92.97% | 319 | *93.31%* | *461* | 93.30% | 478 |
| $\Delta$ | | | -0.23% | -35.43% | *+0.11%* | *-6.68%* | +0.10% | -3.24% |
| Two-Path (-20%) | 87.90% | 116 | 88.63% | 106 | 88.31% | 93 | 88.68% | 94 |
| $\Delta$ | | | +0.73% | -8.62% | +0.41% | -19.83% | +0.78% | -18.97% |
| Two-Path (-100%) | | | 88.63% | 106 | **89.16%** | **48** | *88.94%* | *91* |
| $\Delta$ | | | +0.73% | -8.62% | **+1.26%** | **-58.62%** | *+1.04%* | *-21.55%* |

architectures. For example, when no FLOPs constraint is imposed, the operation-level mutation will find HiAML and NB-101 architectures that remove enough convolution nodes to reduce the model size by more than 85%. These changes drastically reduce the accuracy by over 7.5% on HiAML.

### A.7 AutoGO Components Evaluation

We evaluate the search efficiency on the benchmark families by measuring the speed of each component. The time to execute the search largely depends on the choice of input architecture, i.e., architectures with more nodes and complex topologies like Inception form large search spaces. On the HiAML and NB-201 families, it takes 15 minutes on average to execute a search iteration using the PSC predictor and segment-level mutation. AutoGO visits over 1000 unique architectures per iteration and can find high-performance architectures in around an hour or less.

Specifically, it takes around 1.5 to 2 minutes to segment a parent architecture using BPE, select source and replacement segments, perform resolution propagation, and rank the mutations using the predictor. The bulk of this time is spent between searching the database for replacement segments, confirming their validity and measuring the performance of each mutation, while the BPE segmentation and source segment selection processes take less than 1 millisecond each. When gauging execution time, we sequentially mutate each parent architecture per iteration, but note that this process can be sped up with parallelization.

Resolution propagation with MILP takes 0.11 seconds on average to find a solution or determine that the problem is infeasible. We compare it to an exhaustive search approach by enumerating all candidate solutions. It takes, on average, 0.4 seconds to find a solution and more than 4 seconds for infeasible solutions. Our subgraph extraction process for generating the segment vocabulary is very efficient as the BPE operates on a sequence representation of the CGs. It takes less than 20 minutes to sort all CG topologically, and extract subsequences with BPE.

To provide specific examples of the search time, consider the ResNet-50 Arch 2 and EDSR Arch 3 architectures from Tables 3 and 4, respectively. Mutating the initial ResNet-50 and EDSR CGs takes 1.8 and 1.5 minutes, respectively, on our hardware. It takes longer to mutate ResNet-50 simply because the CG contains more nodes (108) than EDSR, whose CG only has 67 nodes. Moreover, since the base EDSR architecture only uses Convolutions and ReLU operations, we exclude segments

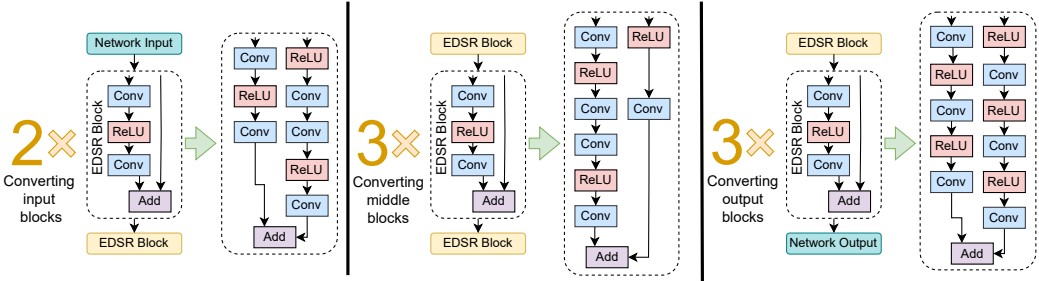

Figure 9: Example mutations performed by AutoGO to create EDSR Arch 2 in Table 4 by swapping out 8 EDSR blocks. Specifically, AutoGO will swap out multiple, simple 'Conv-ReLU-Conv' residual blocks for larger blocks that have operations on both branches.

that contain batchnorm and pooling operations, which reduces the number of replacement segments to consider during mutation.

The first iteration of AutoGO mutates the initial architecture while all subsequent iterations mutate 10 parent architectures. Given that ResNet-50 Arch 2 was found in iteration 3, it took AutoGO around

$$1.8\text{min} + 2\text{iter} * 10\text{arch/iter} * 1.8\text{min/arch} = 37.8\text{min}$$

to discover that architecture. Likewise, EDSR Arch 3 was found in iteration 5, which took

$$1.5\text{min} + 4\text{iter} * 10\text{arch/iter} * 1.5\text{min/arch} = 61.5\text{min}$$

to find. Finally, we note that these measurements and calculations assume sequential processing of parent architectures. In practice (e.g., runtime numbers in Sec 4.2), we use multi-processing techniques to mutate multiple parent architectures simultaneously to further speedup the process.

## A.8 EDSR Mutation Example

Figure 9 illustrates three distinct mutations that take place to produce an EDSR AutoGO architecture. Initially, the EDSR backbone contains 16 'Conv-ReLU-Conv' residual blocks. To create the mutant network, AutoGO removed 8 of these blocks, denoting half the backbone structure, and replaced them with three double-branch structures that also consist of just convolutions and ReLU activations.

## A.9 Hardware and Software Setup

We run our experiments on rack servers using Intel Xeon Gold 6140 CPUs. Each server is equipped with 8 NVIDIA V100 32GB GPUs and 756GB RAM. We execute our search and experiments on Python 3 using `PyTorch==1.8.1` and `TensorFlow==1.15.0`. We implement our predictors using `PyTorch-Geometric==1.7.1`. We use SentencePiece [34] to perform BPE. Finally, we implement our MILP using a Coin-CBC solver [18] and `pyomo==6.4.0` [23].

