# OpenReview forum: "AutoGO: Automated Computation Graph Optimization for Neural Network Evolution"
_NeurIPS.cc/2023/Conference — NeurIPS 2023 poster_

### Official Review · Reviewer_AwFS · 2023-06-25

**Soundness:** 1 poor
**Presentation:** 2 fair
**Contribution:** 1 poor
**Rating:** 3
**Confidence:** 5

**Summary:**

This paper proposed a new NAS algorithm where a performance predictor is built for acceleration. In addition, the experiments are conducted for verification.
============================
Thanks for the authors' rebuttal. Unfortunately, my concerns are still not addressed. For example, 1) the used data are not a benchmark, the reference also did not say it is a benchmark, while the authors believed it is a benchmark. I still believe the novelty of the work is very limited to the community, and cannot address the real concerns in the community.

**Strengths:**

The search space is built on the segments of CG of DNNs.

**Weaknesses:**

Currently, "performance predictor" is more common than "neural predictor," though they refer to the same thing.

Some claims about neural predictors are not correct. For example, "Neural predictors treat NAS benchmarks as datasets." In fact, the early works on this aspect did not use the NAS benchmarks as the dataset, and even these works were earlier than the NAS benchmarks. A baseline in this topic is [1], which is often compared with peer competitors.

[1]  Sun et al., "Surrogate-assisted evolutionary deep learning using an end-to-end random forest-based performance predictor," TEVC 2020.

The predictor is built on the subgraphs minded on NAS benchmarks. To this end, the constructed predictor cannot be generalized to other search spaces and can only be used for the same search space. As a result, the novelty of the work is limited.

The whole algorithm is very similar to the existing NAS algorithms, while the difference is that existing NAS algorithms use the search space composed of architecture units, while the proposed algorithm is based on the segments of CG of a particular architecture. While the motivation is not clear, why the use of segments of CG is more suitable?

The adopted optimization is indeed a multi-objective evolutionary algorithm, while it is called "A Pareto front evolution strategy." In this case, the two objectives, i.e., accuracy and a chosen hardware-friendliness, are treated as conflicting objectives? If it is so, I would ask about the contribution of this work compared to the NAS algorithms falling into the multi-objective NAS algorithms, such as NSGA-NET.

For the example given: "a Conv 3x3 node with incoming edges from Add and BatchNorm operations and an outgoing edge to a ReLU operation as "conv2d3,in,add,batchnorm,out,relu".",  the information of kernel size and stride size have been removed. This involves the encoding of architecture, and there are multiple works in this aspect. Different architecture encodings have different impacts to the performance. Clearly, the one proposed in this paper lost information regarding the architecture.

The experiments on NAS-Benchmarks are not necessary enough because their search spaces are too simple. I suggest the authors check recent works on performance predictors.

**Questions:**

N/A

**Limitations:**

This paper ignores many more existing works on performance predictors, including the encoding of architectures in this topic. Compared to these existing works, the proposed algorithm in this paper has a very limit contribution.

---

> ### Author Rebuttal · Authors · 2023-08-08
>
> ### W1 "Some claims about neural predictors are not correct."
>
> Earlier works like [1] didn't use the term “NAS Benchmarks” as they were published prior to NAS-Bench-101 (which popularized the term). However, [1]'s Introduction says “The training data of the random forest are a set of data pairs, and each pair is composed of the CNN architecture and its performance.” This means the training data is still a benchmark set consisting of labeled architectures. Since 2020 benchmark has become common and usually refers to architectures labeled for performance. Our idea is to pretrain a predictor based on diverse enough families of labeled architectures (called benchmarks) so it generalizes well to other CNNs under generic CG representation.
>
> ### W2 The predictor is built on the subgraphs mined on NAS benchmarks and cannot be generalized to other search spaces.
>
> This is not true. The input to our PSC predictor are subgraphs of available CGs which provide a general architecture encoding. We can achieve generalization because the segments consist of primitive operations (nodes), e.g., Conv
> (torch.nn.Conv2d; tf.keras.layers.Conv2D), BN and ReLU, which are the smallest building blocks of CV models in general. Moreover, our segments are mined via BPE, which will first capture all possible single nodes (smallest granularity is one op) before considering any 2-node, 3-node,... subgraphs.  We train the predictor on a diverse range of NAS benchmarks (80% of 21k architectures) to achieve generalization to general form of CNNs.
>
> In Tables 3-6, we showed that our predictor is generalizable to outside of the NAS Benchmark search spaces it is trained on. We use AutoGO with our PSC predictor to optimize architectures outside of NAS-Benchmarks like ResNets/VGG/EDSR/FSRCNN and even on other tasks. This is exactly in contrast to current predictors in the literature which use encodings to specific to their own cell/backbone design space and cannot generalize.
>
> ### W3  Why the use of segments of CG is more suitable?
>
> First, existing NAS usually defines a search space and searches in that space, e.g., like most other works, [1] uses predefined macro skeletons and op sequences, and only considers a handful of blocks from classical networks like ResNet/DenseNet and pooling blocks and allocate them into a fixed macro skeleton. AutoGO aims to do NAS in a different way, it directly edits a given network's underlying CG (which unifies any network representation) in fine-to-coarse granularity to help ML engineers to fast mutate and deploy a famous network on devices. Our data-driven and algorithm-mined segments replace manual design choices and provide flexibility to mutate a network by single ops or by up to 15-node complex subgraphs (leading to more aggressive topology changes), e.g., Fig. 8 (Supp. Section A.8) shows how 8 classical EDSR residual blocks are replaced by 3 types of bigger blocks with operations on both parallel branches. These bigger blocks are generated using our algorithm-mined segment database and are very hard to be discovered manually for the task.
>
> ### W4  the contribution compared to multi-objective NAS like NSGA-NET.
>
> Multi-objective NAS, which represents large literature, is not our contribution here. We just use a simple evolutionary algorithm to update the Pareto front under defined multiple objectives.  The contribution of AutoGO is a full framework that can directly mutate any input CNN (with our repo of ops and subgraphs discovered from data and without reinventing the search space) to yield a better one for the device (lower latency/GFLOPs and higher accuracy). Our results in Tables 3-6 show clear gains in this use case.
>
> While an early work NSGA-NET does not repeat phases (blocks), it still puts a lot of assumptions on search space design: "for computationally tractability, we (NSGA-NET) constrain the search space such that each node in a phase carries the same sequence of operations, i.e. a 3 × 3 convolution followed by batch-normalization and ReLU."  This is incompatible with NB-201 that use a ReLU-Conv-BN ordering, or EDSR which argues against the use of BN for Super Resolution altogether. Using CG, AutoGO accommodates all of these cases. Another weakness the bit string encoding in NSGA-NET loses information from original graph when mutation and crossover are done on the bit strings.
>
> ### W5 the encoding proposed in this paper lost information regarding the architecture.
>
> The quoted sentence is not related to encoding at all. To clarify, the quoted sentence is only applicable to the segmentation process (a specific step when we convert the CG into a sequence before BPE segmentation). After segmentation, we then map the character-level segmentation back into the CG representation to partition them into subgraphs. In other words, our predictor is based on GNN trained on CGs directly, and AutoGO mutates directly on CGs (graphs) using mined segment database, which keeps all the kernel size, stride and input/output size and other information. Since this information is not lost we can do resolution propagation. It is a strength of this work that mutation is done on CGs directly (rather than on sequential encodings) without losing any information.
>
> ### W6 The experiments on NAS-Benchmarks are not necessary enough because their search spaces are too simple.
>
> First, our experiments are not limited to NAS-Benchmarks. Tables 3-6 shows AutoGO can generalize and optimize a range of real-world CNN architectures (large or small) on diverse tasks/datasets. The goal of Table 2 is to show that AutoGO can even further optimize the “best” architectures in NAS each benchmark, which is something prior literature cannot achieve. Also, it shows that searching on manually designed search space is always not good enough, suggesting the necessity of a data-driven approach like AutoGO to directly edit the CG.
>
> Based on these clarifications we hope the reviewer could reassess the contribution of the paper.

---

> ### Author Response · Authors · 2023-08-21
>
> Dear Reviewer @AwFS,
>
> On behalf of all authors, we thank you for your thorough review. We would appreciate it if you leave any comments on our responses.
>
> We have carefully read your review and diligently provided explanations and clarifications that we believe address your concerns. We kindly request that you consider updating your evaluation if our responses have addressed your main concerns. We remain committed to further refining our work. Your feedback is crucial in ensuring the paper's overall quality, and we greatly appreciate your time and expertise in this matter. Thank you.

---

### Official Review · Reviewer_A8eP · 2023-07-03

**Soundness:** 3 good
**Presentation:** 3 good
**Contribution:** 3 good
**Rating:** 7
**Confidence:** 4

**Summary:**

This paper presents the AutoGO framework, which operates directly on the Computation Graph (CG) of a given DNN architecture. It splits the CG into segments and conducts a search process. Through extensive experiments, the paper shows that AutoGO effectively improves the performance of the top architectures in various public architecture benchmarks. Furthermore, AutoGO demonstrates its capability to automatically optimize different types of large CNN architectures and achieve enhanced results in various computer vision tasks.

**Strengths:**

1. The whole framework from tokenization and mutation to estimation is reasonable and technically sound.
2. It operates the complex problem of directly processing the computation graph and verify on several difficult tasks.

**Weaknesses:**

1. The verification for the ImageNet task is missing.
2. The searched model is highly limited in the current segment database since it only contains the benchmark architectures which are not widely used in different tasks.
3. The training of the accuracy estimator highly relies on the existing collected accuracy and model data pairs. And it would be hard to transfer with only limited accuracy numbers for other datasets and other tasks.

**Questions:**

1. Does the node label contain channel information? The definition is mainly about the graph structure and it seems missing the channel information. And in that way, how would the custom framework adapt to the choice of width, depth, and kernel size?
2. What if the Segment Database is based on some widely used architecture including Conv, MLP, and transformers rather than the NASBenchmark architecture? Would the generated architecture still perform better than the current Pareto front?
3. The choice of the search space covers results mainly in small datasets such as CIFAR-10. How do we use these accuracies to train an accuracy estimator in other datasets or even for different tasks?

**Limitations:**

Please refer to the weakness part.

---

> ### Author Rebuttal · Authors · 2023-08-08
>
> ### W1 Verification on ImageNet
>
> We indeed provide evaluation for ImageNet tasks by training ResNet-50/101 and VGG16-BN on ImageNet and then further fine-tune these architectures on Cityscapes (Semantic Segmentation) and MPII (Human Pose Estimation). We report the results in Table 3 and list our training/fine-tuning setup in Supplementary Section A.4.
>
> ###  W2 Searched models are limited as the segment database only contains benchmark architectures
> Our segment database does not contain the benchmark architectures themselves, but rather the frequent subgraph segments extracted from these diverse benchmarks in a data-driven way. The segments range from 1-node primitive operations (e.g., conv, relu, pooling, add, concat, etc.), most of which are universally present in CNN architectures, up to more complex 15-node/16-edge subgraphs (e.g., the HiAML subgraph used in Figure 4) that vary greatly in terms of computations and hardware-friendliness. For instance, ResNet residual blocks consist of the 'Conv-BN-ReLU' sequence which is in our database, as is the Max Pool operation VGG use for downsampling. Our segments also generalize across different CV tasks, like EDSR [1], which do not use BatchNorm for Super Resolution and instead opt for simpler 'Conv-ReLU-Add' sequences which are also present in our database.
>
> ### W3/Q3 The training of the accuracy estimator highly relies on the existing collected accuracy and model data pairs, which is hard to transfer
> In this paper we do not attempt to train a predictor to estimate the exact task and/or dataset performance of a given architecture. Note how we use Spearman's Rank Correlation Coefficient (SRCC) in Table 1 to evaluate our PSC predictor, which only considers the relative rankings of predictor outputs compared to the ground truth. Rather, our PSC predictor is designed to focus on a how specific segment $s$ in an architecture contributes to the overall performance given its position relative to the rest of the architecture (represented by $P$ and $C$). We do this in order to estimate if replacing (mutating) $s$ with a new segment $s^*$ from our database will bring performance benefits. As such, when AutoGO is optimizing an architecture CG, the predictor acts as a 'proxy' where higher predictor outputs correspond to better architectures. In our experiments on multiple CV tasks and datasets, like ImageNet Classification, Semantic Segmentation on Cityscapes, Human Pose Estimation on MPII as well as Super Resolution on DIV2K/Set5/Set15/etc., and even Denoising using a proprietary in-house dataset, the mutated, hardware-friendly architectures found by AutoGO provide superior performance which validates our approach.
>
> ### Q1 Does the node label contain channel information? How would the custom framework adapt to the choice of width, depth, and kernel size?
> Yes, all CG nodes contain input/output height, width, channel (HWC) tensor size information, while nodes with learnable weights (e.g., Conv, Linear) contain weight tensor size (e.g., convolution kernel size) and a bias boolean as node features.
>
> Therefore, when we use AutoGO to optimize a given input architecture (e.g., ResNet, VGG, EDSR, etc.), we know the HWC information of every operation node that comprises said architecture. During mutation our resolution propagation MILP attempts to adapt replacement segments $s^*$ from our database to match the input/output HWC of the predecessor $P$ and successor $C$ of the input architecture. The MILP accomplishes this by tweaking the strides of convolution/pooling nodes in order to adjust HW as well input/output channels of convolution nodes in the replacement segment $s^*$. We do not adjust the kernel size of convolution operations. Note that in some cases the MILP may not find a solution, e.g., if $s$ performs downsampling, yet $s^*$ does not contain any nodes that can perform downsampling (conv or pooling ops) the proposed mutation {$P, s^*, C$} is deemed infeasible.
>
> ###  Q2 What if the Segment Database is based on some widely used architecture including Conv, MLP, and transformers rather than the NASBenchmark architecture?
> Currently, our database spans a wide range of subgraphs/segments that are used in various convolution-based computer vision models. Our framework could be extended to extract subgraphs from Transformer models which can enrich our database to further improve accuracy and hardware-friendliness. This is definitely a direction for future work given the rise of attention-based structures in computer vision architectures.
>
> References:
>
> [1] "Enhanced Deep Residual Networks for Single Image Super-Resolution" - CVPR'17.

---

> > ### Comment · Reviewer_A8eP · 2023-08-17
> > **Post-Rebuttal Comments**
> >
> > I appreciate your responses addressing my concerns. I literally like the idea of automated graph optimization from segment databases although I still doubt the performance predictor cannot give an accurate performance rank. Overall, I would like to remain my current assessment.

---

> > > ### Author Response · Authors · 2023-08-17
> > > **Re: Post-Rebuttal Comments**
> > >
> > > We appreciate the reviewer very much for the post-rebuttal comments and constructive feedback that helps to improve this paper. Our PSC predictor works in the proposed AutoGO type of subgraph mutation for several reasons.
> > >
> > > First, our predictor uses a P-S-C segmentation scheme, which introduces a form of data augmentation where for each single CG in the training set, we can sample multiple P-S-C combinations based on where the chosen segment S is in the given CG. This P-S-C sampling strategy helped to significantly improve the predictor learning and its ranking performance. Table 1 has verified the effectiveness of the PSC predictor in ranking architectures when the ratio of sampled PSC combinations over original CGs exceeds 1:1, while the conventional way, the GNN trained on the same amount of original CGs, failed on more challenging benchmarks like HiAML, Inception and Two-Path.
> > >
> > > Furthermore, the PSC predictor is not evaluated on absolute architecture performance. Rather, the goal of PSC predictor is to determine which segment mutation (S*) will yield performance improvement in its context (the current parent CG). In other words, our predictor is trained to be sensitive to the choice of segments and segment locations relative to the current parent CG, which suits for the need of AutoGO much better than the conventional GNN predictor. Our experimental results have verified the effectiveness of PSC predictor in capturing the gains from segment-based CG mutations when optimizing a range of architectures on real-world CV tasks.
> > >
> > > Finally, the predictor is trained on the CGs of 5 different NAS Benchmarks, covering diverse ops and topology types. Although each individual benchmark may only cover a predefined set of ops or topological constructs, e.g., NB-101 only uses concat at the end of a cell, whereas Inception and Two-Path benchmarks also allow concat elsewhere. As another example, NB-101 uses 'Conv-BN-ReLU' (ResNet-like sequences), while NB-201 uses 'ReLU-Conv-BN', allowing us to extract 'ReLU-Conv' as a 2-node segment which is useful to EDSR optimization (no BN). In other words, by working with diverse enough benchmark sets to cover a wide range of distinct topological characteristics, we have not only constructed a useful and diverse segment database for AutoGO, but also applied a data science approach to learn from the benefits of segment mutations.

---

### Official Review · Reviewer_sA4y · 2023-07-06

**Soundness:** 4 excellent
**Presentation:** 4 excellent
**Contribution:** 4 excellent
**Rating:** 7
**Confidence:** 4

**Summary:**

This paper introduces AutoGO, an innovative method for evolving neural networks that addresses the challenges of efficiency, low power consumption, and hardware compatibility. AutoGO represents deep neural networks (DNNs) as computational graphs (CGs) comprised of low-level primitives and employs an evolutionary segment mutation algorithm. Notably, AutoGO employs subgraph mining from CGs while utilizing efficient tokenization through Byte Pair Encoding (BPE) from natural language processing (NLP) instead of Weisfeiler-Lehman (WL) kernels. For the evolutionary mutation process, AutoGO leverages neural prediction to explicitly consider positional and contextual information when replacing segments within a CG.

The experimental results demonstrate that AutoGO performs exceptionally well on NAS benchmarks and exhibits promising applications in various domains, including classification, semantic segmentation, human pose estimation, and super resolution.

In summary, this paper presents a novel and effective approach. The writing style is particularly engaging, making it a pleasure to read. I recommend accepting this paper.

**Strengths:**


1. The motivation behind this work is excellently articulated, providing a clear understanding of the research objectives and driving factors.

2. The paper effectively describes recent works in the field, highlighting their significance and comparing their novelty to the proposed method. The related work section is comprehensive and enjoyable to read, showcasing a thorough understanding of the existing literature.

3. This method is technically robust, demonstrating impressive performance across various evaluations. The experimental results validate its effectiveness and reliability, further strengthening the credibility of the approach.

**Weaknesses:**


1. Although the method presented in the paper is highly technical, the focus seems to be predominantly on the technical details rather than providing a comprehensive analysis and intuitive explanations. While I acknowledge the complexity of the method, I believe that enhancing the final manuscript with more analytical insights and intuitive discussions would significantly improve its overall quality.

2. Consider including a simplified pseudocode or algorithmic representation of the method. This would greatly facilitate the understanding of the algorithmic steps and enhance clarity for readers. A concise and structured representation of the method's flow would be beneficial in aiding comprehension.

3. I recommend revising Figure 1 to present the information in a horizontal format. This adjustment would enhance the visual clarity and make it easier for readers to follow the different components and relationships depicted in the figure.

**Questions:**

1. Can the mutation algorithm effectively handle the computational demands of scaling up to large computation graphs? I'm curious to know if it can cope with the complexities involved in processing massive graphs efficiently.

2. When it comes to modeling PSC, does employing a more advanced graph neural network like Graph Transformer provide notable advantages? Or is the choice of GNN design less influential in this particular scenario?

3. Is there a possibility to substitute the mutation algorithm with a GNN policy? I'm interested to hear about any experiences or insights regarding the potential applicability of GNN policies in this context.


**Limitations:**

Please provide a limitation section for future researchers!

---

> ### Author Rebuttal · Authors · 2023-08-08
>
> First, we will add some intuitive explanations to the manuscript and more limitation discussions. The current framework effectively solves the AutoGO problem to mutate CNNs for faster inference and hardware-friendly deployment for a range of CV tasks/networks.
>
> Note that we do provide more information on the analytics of our segment database and a comparison with the WL-kernel of [1] for segment extraction. However, due to the page limit, we had to relegate that material to the supplementary while keeping experimental results that demonstrate the effectiveness of AutoGO on a wide range of tasks, networks and hardware-friendliness metrics in the main manuscript.
>
> ### W2/3  Simplified pseudocode and Fig. 1
> Thanks for the feedback. We have provided an algorithm latex float in the PDF attached to the global response and will definitely revise Figure 1 to give it a horizontal focus.
>
> ### Q1 Handling large computation graphs.
> Yes. In terms of CG size, the largest NAS benchmark we consider is Inception, where the average number of nodes per CG is 673 and the largest CG has over 1500 nodes. By contrast, the next largest benchmark is NB-201, whose largest CG has 336 nodes (half the mean number of nodes Inception has), so Inception contains many massive graphs, yet not only are we able to optimize the best Inception CG, we mine common segments across all Inception CGs to form our database. Granted, AutoGO execution time scales with number of nodes, so optimization on Inception does take longer than NB-201. In Supp. Section A.7 we provide a breakdown contrasting execution time for ResNet-50 (larger CG, 1.8min/arch mutation time) vs. EDSR (smaller with 1.5min/arch mutation time).
>
> ### Q2 Modeling PSC using a more advanced GNN like Graph Transformer
> It is possible to use a more advanced, potentially attention-based GNN and see performance improvements which is an exciting avenue for future work. In this paper, we establish a framework that we could build on top by improving its components, such as the PSC predictor, the search algorithm, and the database, which could result in speedup in the whole search process and better performing mutant architectures.
>
> ### Q3 Mutation for a GNN Policy
> It is interesting for future work to consider a GNN policy-based mutation approach that would model the sampling, aggregate nodes based on their importance, and better capture structural information. This could speed up the mutant-selection process and guide the search toward better candidate replacements that positively affect performance.
>
> References:
>
> [1] "Interpretable Neural Architecture Search via Bayesian Optimisation with Weisfeiler-Lehman Kernels" - ICLR 2021.

---

> > ### Comment · Reviewer_sA4y · 2023-08-17
> > **Post rebuttal**
> >
> > Thank you for the rebuttal. I read the rebuttal carefully and kept the score supporting this paper to be accepted.

---

### Official Review · Reviewer_MAXz · 2023-07-07

**Soundness:** 3 good
**Presentation:** 2 fair
**Contribution:** 2 fair
**Rating:** 4
**Confidence:** 4

**Summary:**

The paper proposes to optimise neural networks by exploiting common subgraphs mined from existing NAS benchmarks - this is achieved by building a vocabulary from networks encoded into topologically sorted sequences and using byte-pair encoding (BPE) to obtain common sequences of operations. After that, a given neural network is segmented and different mined segments are considered as replacement for different identified segments in the given network, all done while taking care of shape propagation. Searching for the best replacement segments is done with a variation of a multi-objective evolutionary algorithm which optimizes for Pareto-efficiency using a proposed (GNN-based) PSC predictor.

**Strengths:**

 - generalizing blocks to segments is an interesting and sensible step towards more flexible NAS
 - the proposed system seems technically advanced, taking care of quite a few corner cases in a convincing way (e.g., solving resolution propagation with linear programming)
 - encoding neural networks as sequences and using BPE is an interesting take on representing neural networks (but could be studied in more details)
 - the proposed PSC predictor seems like an interesting variation of the more standard GNN-based predictors
 - experiments are designed to support claims made in the paper (but results are somewhat hard to interpret, see below)
 - the method (at least after all one-time cost) seems fast, finishing within a few of hours at most

**Weaknesses:**

 - Clarity could be improved in certain places
    - "First, benchmarks (...) requires training the new architecture from scratch" - why is this relevant for the presented work?
    - "(...) predictors learn using high-level cell representation (...), In contrast, AutoGO can mutate an architecture (...)" - why are these two things compared to each other? The ability to mutate beyond an original design space (by the way, this is a tricky thing to formally define, I would appreciate an attempt at that) is orthogonal to a predictor's ability to capture spatial information of a network. Many NAS algorithms achieving similar (or even greater) coverage of architecture than the proposed work, e.g., LEMONADE or $\mu$NAS seem particularly relatable since they utilise mutations towards a similar goal as AutoGO (mutating away from the original design).
    - there seem to be some contradictory information presented regarding what operations are used, first we read (line 43): "(...) by evolving its underlying computation graph (CG) using its original primitive operations.", but then (line 58): "A vocabulary of segments are mined from a large number of CGs from several NAS benchmarks", please clarify
    - The provided definition of PSC does not seem to properly cover nodes parallel to $s_i$.
    - If $s*$ has more than one input, how does the method handle assigning P's outputs to a replacement's inputs? (and analogously for S) Is it a part of the LP problem?
    - it is unclear if a randomly initialized or a pretrained architecture is expected; I couldn't find any information about pretraining a network, but then line 225 says "we retrain all the segment replacements" suggesting the original segments might be trained already (?); it is also unclear why this retraining of segments is needed, considering a performance predictor is used, and how it is done
 - there is some overlap between the proposed method and blockwise NAS works, such as DNA, DONNA or LANA; I think it would be better if the authors acknowledged existence of this line of work, right now it is completely ignored, despite high-level similarities
 - I think the authors should discuss in more details their choice of using toposort+BPE to mine for subgraphs - this approach is bound to fail to recognize many isomorphic subgraphs as the same segments (hinted at the beginning of Section 4), why do you think this is not a problem? How does this greedy approach compare to other alternatives?
 - Results are, generally speaking, hard to compare to the rest of the literature. More specifically:
    - apart from the common benchmarks (NB101, NB201), the paper uses HiAML, Inception and Two-Path - to the best of my knowledge, these are only used by a single, very recent (AAAI'23) paper; however, I don't see any benefits stemming from this choice while it does make comparison to other works harder
    - FSRCNN and U-Net experiments use proprietary networks and tools and, on top of that, only relative improvements are reported in some cases; rendering these experiments basically unverifiable and unusable by the community
 - at the same time, some of the reported results are not particularly convincing, such as:
    - baseline EDSR 2x upscaling performance is actually significantly worse than reported in the original 2017 (!) paper, $\Delta$ PSNR of: -1.25, -1.35, -0.93 and -3.79 for Set5, Set14, B100 and Urban100, respectively. Results are better for DIV2k (I have to assume the authors mean DIV2k validation set), but that's just one dataset out of 5 (and the one used for training),
    - the proprietary FSRCNN also achieves significantly worse results than its parent model, while requiring approx. 150x more FLOPS (!!!)
    - ResNet-50 and ResNet-101 baselines are also worse than reported in the original paper (ImageNet), not to mention any recent improved training recipes
 - it is actually not very clear, but following on the information presented in Section 4.1, it appears that the results in Table 2 were obtained by using a predictor pretrained on 80% of the data available for each benchmark (at least that's the only information we are presented about predictor training, so I'm assuming that's the case for subsequent sections as well); this means that the improvements presented are actually occupied by a very high, hidden cost of having lots of in-domain training data, while in many cases they are not significant
 - perhaps I missed that, but I couldn't find information about the cost of all the pretraining etc.


**Questions:**

See weaknesses, plus a bonus question:  is there a technical reason to stick to (Chinese) characters when representing neural networks? It seems like the method could easily work with just arbitrary numbers instead, so I'm wondering if I missed something or if that's just an arbitrary decision

Also, some typos:

  - line 49: replace ":" with "."?
 - line 160: change to just "AutoGO uses D"? Right now this parts reads like "AutoGO uses D according to D" (since earlier it is said that "segment dataset" == "D")

**Limitations:**

No discussion about limitations - the presentation is actually quite one-sided, by only considering benefits of the proposed method. For example, using BPE is said to "bring several benefits" over methods like WL but not a word about possible downsides (worse handling of isomorphic graphs).

---

> ### Author Rebuttal · Authors · 2023-08-08
>
> ### W1 Comparison to LEMONADE/uNAS/Blockwise NAS
> AutoGO differs from these works. LEMONADE uses network morphism (3.1) rules “Inserting a Conv-BatchNorm-ReLU block”. AutoGO uses diverse, data-driven subgraphs of up to 15 nodes for mutation. uNAS predefines a traditional macro NAS structure in Table 1.
> Blockwise NAS work in a predefined backbone structure, e.g., Fig. 2 of DONNA shows a defined search space over N blocks. DNA and LANA are both distillation teacher-student NAS frameworks that adjust layers/channels per block. The goal of AutoGO as a novel automated toolchain for ML engineers is to eliminate the predefined structure in NAS and to automatically edit the CG of an input architecture with fine-to-coarse segment vocab. It automates the inference acceleration of an architecture on the target hardware without reinventing or limiting the backbone to a manual choice.
>
> ### W2 PSC Parallel Segments/Multiple Inputs
> This is handled by topological sort applied prior to BPE. Numerical labels are assigned to each node, indicating its position in a sequence. Nodes/segments parallel to S would be allocated to P or C. See Figure 6(b) in the supplementary materials and Figure 9 in our global response PDF. Also, if there are multiple valid mappings from P to S (or S to C), then AutoGO chooses one at random.
>
> ### W3 Toposort+BPE Isomorphisms/Comparisons
> The choice of toposort+BPE is a tradeoff between result completeness (extract all subgraphs) and accuracy (estimate subgraph count) for efficiency (memory). While relaxing the problem into mining segments from sequences instead of graphs would hinder recognizing isomorphisms, it enables extracting subgraphs from a multiple CG families and overcome memory inefficiencies. By converting back the extracted subsequences into their original subgraph form, we are able to recognize isomorphism among extracted subgraphs. We compare to WL-Kernel and show the speedup of our approach compared to [1], who extract motifs. However, WL-kernel has disadvantages. Primarily, it does not scale well with the size of the graph in terms of RAM and execution time. This limited the number of CGs we could consider at a time so we couldn’t perform extraction over all CGs in a NAS benchmark dataset (we can with BPE). Even on a subset of graphs extraction took at least 6 hours (BPE takes less than half the time). We describe this comparison and provide a figure in Supp. Section A.3.
>
> ### W4 Results hard to compare to the rest of the literature; use of newer benchmarks
> 1.	Prior literature finds architectures amongst a predefined search space. In Sec. 4.2, we show that even the best architectures from 5 existing NAS benchmarks can be edited by AutoGO for improved accuracy and lower FLOPS, showing the bottleneck in NAS is the manually designed search space in the first place, not the search algorithm. Therefore, the AutoGO concept is new; it can edit and mutate an existing architecture for better accuracy/faster inference, an ability that the prior works lack because they would define their own search space of backbones/blocks, which limits performance.
> 2.	Using newer benchmarks increases the diversity/coverage of our segment database, which brings benefits to downstream tasks, e.g., Figure 4, where a HiAML block is adopted by a mutant of ResNet-50 by AutoGO.
>
> ### W5 Comparing Tables 3-5 to the original papers
> We want to clarify that rather than advancing the training performance of EDSR, ResNet50/101, the goal of experiments in 4.3 (Tabs. 3, 4)  is to show that even on established neural architectures and on GPUs, AutoGO can still automatically optimize these architectures for more efficient computation. This ability of automatically generating better architectures based off original architectures is no coincidence and is verified through extensive experiments on multiple networks.
>
> The performance metrics we report are not directly comparable to the top-1 or PSNR numbers reported in the original papers for two reasons: To provide a fair comparison between our baseline architectures and the mutants optimized by AutoGO, we use a function provided by the CG API to instantiate a trainable model from the CG (as the mutant architectures returned from AutoGO use this format) which results in some implementation differences between the CG-instantiated model and the model definition in the original source code (as the CG API needs to be general enough to support many architectures). Second, our training hyperparameters (Supp. Section A.4) differ from the original papers, e.g., the original EDSR paper uses an input patch size of 48 and halves their learning rate at specific steps. We use a patch size of 64, a cosine scheduler, and train for less time.
>
> These changes produce results which are not directly comparable to what is originally reported but not always worse, e.g., our baseline VGG16 exceeds the VGG16-BN result reported in torchvision (74.18% vs. 73.36%) and our AutoGO optimized architecture further improves this to 74.91%. Also, all of our CIFAR-10 results in Table 2 use the same training setup, and our baseline performance is higher on NB101 (94.24% vs. 95.18%) than what is originally reported.
>
> ### Bonus Question
> BPE works best when applied to sequences of atomic characters, e.g., a single character like ‘A’, '1' or ‘壹’ represents a single meaningful entity (one node in the case of CGs). We originally tried representing nodes with English characters but found there were not enough of them given our “node|incoming|outgoing” format, so we simply switched to Unicode Chinese characters.
>
> References:
>
> [1] "Interpretable Neural Architecture Search via Bayesian Optimisation with Weisfeiler-Lehman Kernels" - ICLR 2021.

---

> ### Author Response · Authors · 2023-08-21
>
> Dear Reviewer @MAXz,
>
> On behalf of all authors, we thank you for your thorough review. We would appreciate it if you leave any comments on our responses.
>
> We have carefully read your review and diligently provided explanations and clarifications that we believe address your concerns. We kindly request that you consider updating your evaluation if our responses have addressed your main concerns. We remain committed to further refining our work. Your feedback is crucial in ensuring the paper's overall quality, and we greatly appreciate your time and expertise in this matter. Thank you.

---

> > ### Comment · Reviewer_MAXz · 2023-08-21
> > **Reply**
> >
> > Thank you for your rebuttal and please accept my apologies for a late reply.
> >
> > While usually I try to comment on each point individually, considering the amount of points raised by the authors I will try to summarise my thoughts in a more concise manner for the sake of brevity.
> >
> > On the high-level, my concerns about comparability and clarity remain.
> >
> > Regarding the former, I understand that the authors aim to improve networks and not beat SOTA, so I'm not asking for the best possible numbers in the absolute sense. My concerns are based on the fact that it is usually easier to optimise a network that is trained suboptimally. Being able to Pareto-dominate a baseline is the golden target for any optimisation work, but it is not worth that much if the baseline is unable to reproduce 8 years old results (ResNet). Also, I'm going to ignore VGG because this model is so old and suboptimal (in the sense of its architecture) that it really should not be used anymore - improving upon it is a really low bar.
> > What is more, we have to remember that comparability issues work in more than one way - even if we agree that the results presented in the paper are enough to convincingly assert strong performance of the proposed method, how is that going to work for potential follow up works? Comparison to FSRCNN and UNet is impossible, whereas comparison to any other model would either require reusing the same hyperparameters as in this paper (which are not used by anyone else and often seem to produce suboptimal results) or retrain models reported here in a new setting. In either way it seems the burden of comparing to this paper would be unfairly shifted onto the follow up works, which is not desired.
> >
> > For the latter, I feel like the authors often mean something different than what is written. For example, in the rebuttal we can read "[AutoGO] can edit and mutate an existing architecture for better accuracy/faster inference, an ability that the prior works lack because they would define their own search space of backbones/blocks, which limits performance." I genuinely hope the authors did not mean to say their work is the first one that proposes to mutate architectures? In general, my understanding is the novelty of this work lies in the automated extraction of the segments library (and also some improvements to the predictor etc. but that's not relevant for my point now) - all the other things frequently mentioned by the authors, like the ability to mutate etc., are simply derivatives of this aspect and on their own are not qualitatively new.
> > Also, the authors mention on a number of occasions that the baseline method are limited because they require a predefined search space (e.g., pt. 1 in W4). Again, this is a bit misleading. Every method requires well-defined boundaries in order to be implemented - this is also the case for the proposed method (and is the source of my question regarding what operations are used, see a bullet point below). The limiting factor is not the existence of a predefined search space on its own but its size - the proposed method has the advantage of being able to span more networks designs easily (e.g., due to the automated mining of segments) but fundamentally it also utilises a predefined search space, even if defined implicitly.
> >
> > Some notable specific comments:
> >
> >  - my question about the overall cost has not been answered
> >  - my question about what operations are used has not been really answered - I understand the overall set of operations used is summarised in bottom right plot in Figure 5? Although we know from footnote 1 that "conv" exists in at least a couple of different variations so the information in the appendix is incomplete. Also, my comment was about contradictory information in the text - as far as I understand "using its original primitive operations" should be rather "using operations extracted from NAS benchmark". The point is: after learning BPEs from benchmarks, if I wanted to optimise a network that has completely different operations (e.g., a transformer) - will mutations be able to use operations from the model (not present in the segment database) or not? One part of the paper suggests it can happen, the other the opposite.
> >  - my question about the amount of labelled data needed to train the proposed predictor has not been answered
> >  - the authors say the choice to use BPE over WL is a trade-off, but they never discuss the downside of using BPEs, even after pointing this out in the review; while I understand it might not be feasible to scale WL to the level needed to run experiments, considering the authors already present comparison in terms of time (which is a win for BPE) I don't see a reason why not to include at least some approximated comparison on the other side of the trade-off (completeness)
> >
> > With all this in mind, I would consider my current score to reflect my opinion well - the submitted paper is solid but overall the shortcomings make me lean towards rejection.

---

> > > ### Author Response · Authors · 2023-08-22
> > > **Re: reply**
> > >
> > > We would first like to thank the reviewer very much for taking the time to read our rebuttal and posting thoughtful comments.
> > >
> > > Regarding the comparability issue, AutoGO is an automatic framework aimed at optimizing a given neural architecture for hardware-friendly inference. The purpose of the experiments is to demonstrate that we are able to incrementally improve a given CNN architecture toward lower latency/FLOPS/power with maintained or higher performance. For all the experiments, the original network and the AutoGO-improved network are trained on the same training recipe. The contribution lies in a Computation Graph optimization framework that does not require users to manually specify their search space like NAS would do. The framework also includes the necessary components for AutoGO to operate and function on CGs, including PSC-predictor, segment extraction method, and resolution propagation, and a maintained segment database for CG mutations. This is a novel contribution on the end-to-end system level for inference acceleration (rather than optimizing training) compared to those prior NAS works that search on expert-defined search spaces. Significant efforts were involved in developing AutoGO and demonstrating benefits on a range of representative neural networks and CV tasks, which was actually not easy to achieve.
> > >
> > > Reviewer: “Even if we agree that results presented in the paper are enough to convincingly assert strong performance of the proposed method, how is that going to work for potential follow up works?”
> > >
> > > The fact that AutoGO-optimized architectures outperform original architectures can be verified by comparing the architectures and training them under the same training recipe, which we did. Also, we submitted the experimental code and commit to open-sourcing the research code including our algorithms, segment database, architectures before and after applying AutoGO, which are valuable and beneficial to further studies on automated computation graph optimization and on-device DNN acceleration/deployment in real-world scenarios.
> > >
> > > Regarding the clarity issue, please allow us to clarify that the rebuttal could contain more oral language (that is not placed in the full context) than what is in the paper for explanation purposes. While we understand what the reviewer means, we would kindly invite the reviewer to refer to the paper for more formal claims regarding our technical contributions and motivation. And this sentence from the rebuttal doesn’t mean we are the first to propose mutations to neural architectures. However, in terms of the usage in reality, AutoGO is the first of its kind, it’s a framework that can automatically edit any user-supplied CNN without requiring the user to specify the search space manually, i.e., defining cells, blocks, or backbone structures, which is an arguably onerous job required by prior NAS methods. Admittedly, we still perform search within the search space implicitly induced by our segment database for Computation Graph mutation (a point we fully agree with the reviewer). Thank you for pointing it out!
> > >
> > > Reviewer: "The limiting factor is not the existence of a predefined search space on its own but its size - the proposed method has the advantage of being able to span more networks designs easily (e.g., due to the automated mining of segments) but fundamentally it also utilises a predefined search space, even if defined implicitly."
> > >
> > > We definitely agree with reviewer: our algorithm-mined segment database has a coverage of diverse types and sizes of convolutional ops and structures. AutoGO is the first automated solution to editing CNNs on this induced CG search space which provides more flexibility than each manual design. We will further polish introduction, related work, conclusion to make sure this point is clarified.
> > >
> > > Our work offers a novel angle to search space design based on data science. We show that by mining a sufficiently diverse set of NAS Benchmarks (5 here), it is feasible to apply a data science approach to CG mutation space construction to replace human-defined block construction or expert rules in NAS. We are trying to promote an alternative idea for future NAS: mining and maintaining a shared subgraph repo to evolve low-level CGs efficiently for AI deployment, instead of re-defining search spaces for the task and for device, which involves trial-and-errors and limits the application of NAS to AI deployment, especially given the rise of AI-running edge devices (as our real-world FSRCNN/Denoising UNet results are targeting). We included these results in Sec. 4.5 and do not deem them as a weakness, since these proprietary networks show the real-world use cases for AutoGO in IoT in addition to other results. And we believe our extracted segment database, which we commit to open-sourcing and maintaining constantly, is relevant and valuable to the research community for future studies on automated DNN acceleration and deployment in practice.

---

> > > ### Author Response · Authors · 2023-08-22
> > > **Addressing Some Specific Concerns**
> > >
> > > Responses to specific questions/concerns:
> > >
> > > ### Q1/3 Overall cost and amount of labelled data for predictor training:
> > > We apologize for the negligence that answers to these questions were not properly posted in the first rebuttal although we tried to. The predictor training was not costly. To train/test the PSC predictor used throughout the paper, we use 21k labelled architecture CGs (including NB-101: 5k randomly selected CGs, which is ~1.2% of the entire NB-101; NB-201: 4096 archs or 26.2%; HiAML: 4.6k, Inception: 580; Two-Path: 6.9k). These settings and numbers are listed in Table 7 in the paper. Each CG can be decomposed into many P-S-C data samples taken from the original CGs, e.g., in Table 7 the 5k NB-101 samples become over 400k P-S-C samples but there are still at most 5k unique accuracy labels. Then, we only need 17k (80%) of the labelled architecture CGs to train the PSC predictor used in generating results for Tables 2-6. Another 2k (10%) are used in the validation set but the predictor doesn't learn from them, while we use the last 2k (10%) to evaluate the predictor in terms of SRCC in Table 1. The same predictor trained in the above way is used generate other results throughout the paper (including Tables 2-6).
> > >
> > > The predictor training process took around a day on our hardware (described in Supp. Section A.9) and we note that much of this time is due to I/O loading/bandwidth (which a dedicated software engineer could optimize) as the PSC samples are stored in large caches. Overall, the predictor training on 80% of 21k labelled architectures was not costly.
> > >
> > > ### Q2 Operations used:
> > > The bottom right histogram in Figure 5 provided overall statistics, which group all operations by the primitive name, e.g., “conv“ represents all convolution variations together regardless of kernel sizes which are {1, 3, 5}. The exception is depthwise convs which have kernel sizes of {3, 5}. We can also provide more detailed breakdowns of statistics if needed.
> > >
> > > In this paper, we focus on implementing a working AutoGO framework for CNNs for CVs tasks. Our segment database is extracted from diverse enough NAS Benchmarks which covers a wide range of primitive operations which compose CNN architectures (not Transformers). However, we will refine the statement into (we mutate) “using operations extracted from NAS benchmarks” rather than mutating “using its original primitive operations” as suggested by the reviewer.  In AutoGO, it is true that each mutation must use ops that appear in the segment database already. Thanks for pointing it out. The reason we said mutating “using its original primitive operations” was because here we focus only on CNN primitive ops which our extracted segment database covered as single-node segments (i.e., primitive ops). But we will certainly refine these statements taking the reviewer's suggestion into account.
> > >
> > > ### Q4 BPE limitations:
> > > In the rebuttal reply to "W3" we mention that the choice of toposort+BPE is explicitly a trade-off between completeness and efficiency and state that while BPE is much faster (compared to WL-Kernel subgraph mining, which is actually infeasible to be executed on the CGs of all 5 benchmarks on which we performed segment extraction), its limitation is to handle isomorphisms. But this is not a big issue, since we can remove redundant segments, i.e., the isomorphic subgraphs that have different sequential encodings by toposort+BPE, in order to yield only unique subgraphs (segments), as mentioned in line 259. This filtering was fast, since we only needed to compare subgraphs that have the same number of nodes to check for any presence of isomorphism.
> > >
> > > In BPE algorithm, the vocabulary size will critically determine how many subgraphs we can extract and the associated cost. In this paper, we strike a balance by setting vocabulary size to 2000 which results in subgraphs up to 16 nodes and edges. The segment database constructed this way together with the pertained PSC predictor under this segment database (vocabulary) is sufficient to yield benefits on the range of networks and tasks we optimized using AutoGO. However, it's worth noting that the BPE tokenization algorithm works recursively by extracting all 1-node segments, followed by all 2-node segments, and then all 3-node segments, and so on. This means our segment database can grow incrementally which is another advantage: when the current database is not sufficient, we can always try to include more complex subgraphs with more nodes/edges into consideration by further running BPE.
> > >
> > > We would like to appreciate the reviewer's time and efforts to read our responses again and reconsider the evaluation. We hope we have addressed the reviewer's main concerns with these responses. We are committed to further refining our work and making the necessary improvements to address these concerns and any further concerns you may have. Thank you for the opportunity to enhance the quality of our research.

---

### Author Rebuttal · Authors · 2023-08-08

We would like to thank the reviewers for their constructive comments, for the suggested references, and for pointing out several typos, which we will fix. In addition to the individual responses, we are providing a PDF containing some new figures/tables.

We would like to clarify that the motivation of AutoGO is to provide an automated framework to help ML practitioners to optimize larger networks for deployment (with faster/lighter inference) on their target hardware (usually an edge device like cell phones or cameras instead of GPUs), such as FSRCNN on Super Resolution, and a U-Net on Image Denoising to optimize PSNR and latency/power for a mobile NPU. Most existing NAS is devoted to optimizing architectures by proposing a NAS search algorithm to explore the architectures or encodings of architectures in a manually defined search space, either pushing the accuracy to the limit or performing multi-objective search. However, none of the existing works will suit deployment and inference acceleration scenario above for several reasons:
1) When deploying an ML task in reality, ML practitioners seldom resort to NAS, because the manual redesign of a search space is the hardest to decide. This is especially challenging when one has multiple devices or multiple generations of devices, one would want to quickly migrate models across devices. Rather than redesigning everything including the search space from scratch, one will take an existing famous architecture already reported by the literature on that task and start to edit it to fit into the hardware while maintaining accuracy as much as possible. That is exactly the process AutoGO tries to automate, rather than searching for another large network that beats SOTA on a CV task.
2) Given the large body of literature on NAS, the search algorithm (EA, RL, BO), the multi-objective search, and encoding of architectures, are already heavily studied--they are not the bottleneck. The bottleneck is how to come out with the search space for the task. Note that existing scientific research on NAS usually manually designs a search space that is good to boost scores and still friendly to GPUs but not necessarily friendly to other daily-used devices. AutoGO proposes a novel data science approach to this dilemma by mining segments from a diverse range of benchmarks featuring different kinds of topological features and ops and maintain a segment vocabulary. By directly operating on CG level (a fundamental representation of any network), AutoGO can directly edit any given input CNN's CG using this vocabulary (without distorting network representation), thus avoiding confining search to specific, manually designed cell/block structures or macro backbones or to any assumption.  The segment vocabulary contains fine-to-coarse subgraphs which can be constantly updated to supply to AutoGO.
3) We optimize on a range of NAS benchmarks to show that AutoGO can still enhance the best architectures in them, while exhaustive search cannot, because the search space design is not sufficient. We show AutoGO can generalize to enhance high-resolution networks ResNet EDSR VGG etc. in both accuracy and inference speed even on GPU. Finally, we demonstrate the already very lightweight manually designed proprietary FSRCNN and U-Net can still be optimized by AutoGO to achieve low-power/latency inference on cellphone. These results are all challenging to achieve.

Also, we clarify some writing in the paper:
### L91: "First, benchmarks only provide performance annotations for architectures inside a manually designed fixed search space." relevance to this work? (MAXz)

If we change an architecture in NB101 by tweaking ops or channels in a cell, it will result in an architecture outside the predefined NB101 search space, whose performance can't be assessed by a predictor trained on original NB101 encodings. In contrast, our predictor can as it operates in lower-level CG space.

###  What operations are used (MAXz)
We mutate an architecture using BPE-extracted segments, each composed of primitive operations like conv or relu. Segments range from all primitive operations (single nodes) to any 2-node sequences, ..., up to the largest segments in our database (Supp. Section A.1) with 15 nodes.

### Line 225 “Retrain” Typo (MAXz)
This is a typo and is supposed to be “retain”. Mutation is guided by the PSC predictor, without requiring any retraining.

### Our FSRCNN has 150x more FLOPS? (MAXz)
This is not the case. Our Proprietary FSRCNN-3,4 models in Tab. 5 are not comparable to e.g., FSRCNN-7 model in ECBSR paper. First, ours have 3 or 4 convolutional layers, while FSRCNN-7 has 7 layers (6 GFLOPs by Table 1 of the ECBSR paper) and uses a 9x9 Deconv for upsampling. But ours uses a 2x2 Deconv to make it friendly to cellphone hardware. Moreover, the FLOPs reported in Table 5 used an input image size of 64x640 which is what our power profiler allows. Also, there is a typo in Table 5 that FLOPs should be in units of 1e6, not 1e9. To clarify all of these, we provide a new Table 10 in the attached PDF with power metrics and FLOPs for input size 640x360 (which is used in ECBSR paper). At this size, our FSRCNN-3,4 base models have 2.67/3.74 GFLOPs, which is significantly less than the 6 GFLOPs of FSRCNN-7 in the ECBSR paper. Yet, the point shown by Table 5 is that AutoGO can still optimize such small models.

### FSRCNN, U-Net experiments use proprietary networks (MAXz)
AutoGO aims to assist ML engineers to auto-tune a known architecture on their target hardware (e.g., a cellphone) for faster inference. Our last results in 4.5 on FSRCNN, U-Net demonstrate this ability of AutoGO in reality, which complements our other results in 4.1-4.4 on public datasets/networks. We could only post relative changes at submission, but have been cleared to post power/lat metrics. The power values for the Table 6 U-Net are 724.59mW and 657.82mW for the original and AutoGO-optimized models, respectively.

---

> ### Comment · Area_Chair_j9cF · 2023-08-19
>
> Dear Authors,
>
> thank you for the detailed responses to all reviews! We will take these details into consideration for the further discussion.
> @ Reviewers MAXz and AwFS, do you have any further questions or discussion points for the reviewers?
>
> Best regards,
> AC

---

### Decision · Program_Chairs · 2023-09-21

**Decision:**

Accept (poster)

**Comment:**

This paper proposes AutoGO, a NAS framework that operates on the Computation Graph of a given architecture. The CG is partitioned into segments, i.e. it extracts subgraphs from existing benchmarks, to facilitate efficient architecture mutation. A "vocabulary" of segments is extracted from architectures using byte-pair-encoding. The paper empirically shows that AutoGO can improve the performance of the top architectures in common benchmarks. Furthermore, AutoGO demonstrates its capability to automatically optimize different types of large CNN architectures and achieve enhanced results in various computer vision tasks. The best replacement for a segment is searched with a variation of a multi-objective evolutionary algorithm (EA) using a GNN-based predictor.
The proposed model is innovative as it uses the architecture partitioning in an efficient way to allow for efficient EA. The provided experimental validation is extensive. Therefore, two of the reviewers provide a positive rating of the paper. At the same time, there is criticism of the significance of the approach, since it does not clearly improve over the SotA. Details on the pareto-optimality of the optimization have also been discussed, as well as the delineation form prior works on architecture segmentation. Still, I believe this paper proposes a very interesting new approach that might help to advance NAS and multi-objective NAS in the future.